# An inventory of Alpine drought impact reports to explore past droughts in a mountain region

Ruth Stephan[1], Mathilde Erfurt[1], Stefano Terzi[2,3], Maja Žun[4], Boštjan Kristan[5], Klaus Haslinger[6], and Kerstin Stahl[1]

[1]Environmental Hydrological Systems, Faculty of Environment and Natural Resources, University of Freiburg, Freiburg, 79098, Germany
[2]Institute for Earth Observation, Eurac Research, Viale Druso 1, 39100, Bolzano, Italy
[3]United Nations University Institute for Environment and Human Security (UNU-EHS), Platz der Vereinten Nationen 1, 53113 Bonn, Germany
[4]Slovenian Environment Agency, Vojkova 1b, 1000 Ljubljana, Slovenia
[5]Slovene Chamber of Agriculture and *Forestry*, Institute of Agriculture and *Forestry* Maribor, Vinarska ulica 14, 2000 Maribor, Slovenia
[6]Climate Research Department, Central Institute for Meteorology and Geodynamics (ZAMG), Hohe Warte 38, 1190 Vienna, Austria

**Correspondence:** Ruth Stephan (ruth.stephan@hydrology.uni-freiburg.de)

**Abstract.** Drought affects the European Alpine mountain region, despite a humid climate. Droughts' damaging character in the past and an increasing probability in future projections call for an understanding of drought impacts in the mountain regions. The European Drought Impact report Inventory (EDII) collects text reports on negative drought impacts. This study presents a considerably updated EDII focusing on the Alpine region. This first version release of an Alpine Drought Impact report Inventory (EDII$_{\text{ALPS}}$) classifies impact reports into categories covering various affected sectors and enables comparisons of the drought impact characteristics. We analysed the distribution of reported impacts on the spatial, temporal and seasonal scale, and by drought type for soil-moisture drought and hydrological drought. For the spatial analysis, we compared the impact data located in the Alpine region to the whole of Europe. Further, we compared impact data between different climatic and altitudinal domains (Northern vs. Southern region, pre-Alpine vs. high-altitude region), and between the Alpine countries. Compared to the whole of Europe, in the Alpine region agriculture and livestock farming impacts are even more frequently reported, especially in the Southern region. Public water supply is the second most relevant sector, but overall less prominent compared to Europe, especially in spring when snowmelt mitigates water shortages. Impacts occur mostly in summer and early autumn with a delay between those impacts initiated by soil-moisture and those initiated by hydrological drought. The high-altitude region shows this delay the strongest. From 1975 to 2020, the number of archived reports increases, with substantially more impacts noted during the drought events of 1976, 2003, 2015 and 2018. Moreover, reported impacts diversify from agricultural dominance to multi-facetted impact types covering forestry, water quality, industry and so forth. Though EDII$_{\text{ALPS}}$ is biased by reporting behaviour, the region-specific results of negative drought impacts across the water-rich European mountain region demonstrates the need to move from emergency response to prevention and preparedness actions. These may be guided by EDII$_{\text{ALPS}}$' insights to regional patterns, seasons and drought types.

## 1 Introduction

Droughts are natural hazards, which can cause widespread and severe impacts on the environment and societies. Compared to other weather-related hazards, such as floods and storms, droughts are among the most damaging events in terms of affected people and economic costs (Wilhite, 2000a; UNISDR, 2009; UNDRR, 2019). The summer droughts of 2003, 2015 and 2018

have raised concerns about the vulnerability of the European water budget to climate change (Weingartner et al., 2007; Teuling, 2018), because these events have affected more than 17 % of the European population (Mastrotheodoros et al., 2020). Due to the mountain climate with an annual total precipitation between 400 to beyond 3000 mm/year (Isotta et al., 2014) and the four major European rivers Po, Rhone, Rhine, and Danube, the Alps are also called the "Water towers for Europe" (Viviroli et al., 2007). Nonetheless, past droughts caused severe impacts such as limited supply of water for drinking, irrigation and

hydropower generation (Haslinger et al., 2019). The predicted increase of drought frequency, duration and extent stresses the relevance of systematic analysis of drought impacts and their cascading effects in mountainous areas. This is particularly relevant within the Mediterranean climate in the Southern parts of the Alpine regions, where recent drought events triggered water disputes and spread of multiple impacts (Yves et al., 2020). The need to understand the role of drought impacts in Europe's mountainous region is stressed by the fact that more than 170 million people live within the major river basins

(Viviroli et al., 2007). Until now and to the best of our knowledge, only the expert paper by the "Water management in the Alps" platform (Water Management in the Alps Platform of the Alpine Convention, 2018) is a transnational study focusing on drought impacts in the Alpine region presenting experiences, approaches and common challenges for water management by stakeholders. This expert paper emphasizes the need to move from emergency to preparedness actions, essential for the Alpine-wide research on past drought and potential future impacts.

Drought builds up slowly and accumulates over time with cascading effects and the provoked impacts may linger for years after termination (Wilhite, 2000b). Compared to other disasters these characteristics brought up different drought definitions with the difficulties to determine the onset and termination of the phenomenon. A common approach is to define drought as a sustained period of below-normal water availability (Tallaksen and van Lanen, 2004) and classify the phenomenon into different types that generally occur in the following order (Wilhite and Glantz, 1985; van Loon et al., 2016): (1) Climate variability leads

to precipitation deficit causing meteorological drought ($D_M$), the initiator for the other types. Meteorological drought combined with high potential evapotranspiration leads to (2) agricultural or soil-moisture drought ($D_{SM}$). (3) Hydrological droughts ($D_H$) occur delayed, associated with the effects of temperature anomalies, precipitation shortfalls, and/or anthropogenic demand pressures on surface or subsurface water supply (e.g. streams, reservoirs, lakes or groundwater). In a mountain-to-foothill region this propagation may differ as hydrological processes vary from high to low elevations. The annual hydrological cycle

may be more likely to be reset every year by winter snow. In addition, response and reaction times are fast, gradients steep and storages more local and diverse.

The different drought types generally lead to a wide range of impacts, making an impact assessment more difficult compared to other disasters. $D_M$ is typically understood as the prime trigger and $D_M$ impacts may often have compound causes with heat waves in lowlands, e.g. excess mortality as a result of cardiovascular diseases. In mountain regions such direct impacts are likely less relevant due to the cooler climate. Most of these direct drought impacts can be linked to either $D_{SM}$ or $D_H$. For example, low soil moisture typical for $D_{SM}$ initiates reduced vegetation health or crop quality, whereas low discharge and/or groundwater storage typical for $D_H$ causes problems in public water supply (Wilhite and Vanyarkho, 2000). Drought impacts not directly induced by the conditions of $D_{SM}$ or $D_H$, are also called "2nd-level" or "indirect" impacts (Wilhite and Vanyarkho, 2000). Typical examples for such impacts are increased costs due to supplementary irrigation, increased disease attacks on trees or water allocation conflicts. For these impacts drought can be the trigger, but none of the described types can be identified as the exclusive driver. Therefore, they are known to be the the least tangible. In order to link drought impacts specifically to drought types, $D_{SM}$ and $D_H$ are the most evident types (Stagge et al., 2015). Despite the challenge to identify drought impacts, several efforts have been made predominantly focusing on the agricultural sector (Logar and van den Bergh, 2013; Poljanšek et al., 2019; Cammalleri et al., 2020), but not specifically on mountain regions or mountain-to-foothill transitions. Stahl et al. (2012) introduced the European Drought Impact report Inventory (EDII) to widen the perspective to the broad scope of drought impacts including more sectors, such as *Public water supply*, *Tourism and recreation* and *Energy and industry*. The EDII defines drought impacts as negative consequences for environment, society or economy and classifies these into 15 sectoral categories with multiple subtypes. Blauhut et al. (2016) and Stahl et al. (2016) used these geo-referenced reports to compare sectoral differences across Europe, which demonstrates the value of this impact inventory. Stagge et al. (2015) and Bachmair et al. (2016) statistically modelled the likelihood of impacts based on EDII data.

This study builds up on the drought impact data collected and classified in the EDII, expanding it with the help of existing databases to develop a mountain specific "Alpine Drought Impact report Inventory (EDII$_{ALPS}$)". The main objective is to survey, classify and systematically assess past drought impacts in the European Alpine region with the following leading questions:

– How do impacts differ in such a mountain-foreland region compared to the whole of Europe? Are there any spatial patterns within the Alpine region driven by altitudinal or climatic conditions (high-altitude region, pre-Alpine region, Northern and Southern region)?

– Are there any trends of drought impact frequencies over time, overall and for different seasons?

– Are there seasonal differences in the occurrence of impacts related to soil-moisture drought versus impacts related to hydrological drought?

## 2 Methods

### 2.1 The study region and its specific characteristics

Our study region is the so-called "Alpine Space" introduced by the EU-funded programme of the same name (Fig. 1, Interreg -Alpine Space Programme 2014-2020). The Alpine Space covers the Alps and their foothills, as well as different climatic zones

and therefore allows the consideration of water and resource flow and exchange typical to mountain regions. With the study region's extent, we therefore include drought impacts not only at high altitudes, but also in downstream areas of the water-rich source regions (e.g. the river basins Po, Rhine, Danube etc.). We updated the content and analysed impact patterns within subregions of the Alpine Space and compared this new $EDII_{ALPS}$ with the whole pan-European region of the EDII (Stahl et al., 2016) which we name $EDII_{EU}$ in all comparisons (Fig. 1)

The Alpine Space boundary corresponds to the borders of NUTS regions. The Nomenclature of Units for Territorial Statistics (NUTS) is a spatial unit with levels 1, 2 and 3 used in the European Union to subdivide countries into major, middle in minor regions for statistical purposes (Eurostat, 2020). Using these NUTS regions, we split the Alpine Space into two subdivisions in order to compare contrasting climatic and altitudinal conditions: (1) The "Northern region", with all NUTS 2 (and thus including all NUTS 3) regions characterised by maritime climate versus (2) the "Southern region" with the other NUTS regions characterised by the Mediterranean in the South and Southwest and continental climate in the Southeast (Bouma, 2005). Haslinger et al. (2019) presented a strong North-South dipole along the Main Alpine Crest for dry and wet areas during the last 210 years. (3) The "high-altitude region" identified with NUTS 3 regions for which $\geq 30$ % of the area are higher than 1000 masl versus (4) the "pre-Alpine region" covering all remaining NUTS 3 regions. For the altitudinal comparison, we chose NUTS 3 regions with a higher spatial resolution, because altitudinal characteristics do present stronger the smaller the area. In the following the term "domains" includes the pre-Alpine, high-altitude, Southen and Northern region, the Alpine Space covered by the $EDII_{ALPS}$ and Europe covered by the $EDII_{EU}$, and thus differs from smaller spatial units.

## 2.2   Collection and pre-processing of drought impacts data

We retrieved drought impact data from different sources located in the European Alpine countries considerably updating the latest version of the EDII (i.e. status of EDII from September 2019, Stahl et al. (2016), https://doi.org/10.5194/nhess-16-801-2016). The EDII itself archives text-based reports on drought impacts from different sources, most frequently from newspaper articles, web pages, scientific or governmental reports, databases and other information sources. The first archived impact report dates back to 1448. This historical information for southwestern Germany was retrieved from the collaborative research environment tambora.org (Glaser et al. (2015), https://www.tambora.org/). However, most collected reports stem from the late 20th Century till now. For the search of text-reports we applied the same method described in Stahl et al. (2016) and focused on the last 20 years. The compiled impact data had to fulfill the standards the EDII requires in order to be consistent, e.g. impact description, reference, location and timing. Therefore, our impact collection is a substantial update for EDII with a region-specific focus. Subsequently, the collected impact reports come from sources located in the different Alpine-countries: (1) A broad variety of Italian and (2) German text-reports, (3) the French platform Propluvia (Ministère de la Transition Écologique et Solidaire, http://propluvia.developpement-durable.gouv.fr), (4) the Austrian chronicle of severe weather impacts "Unwetterchronik" (Zentralanstalt für Meteorologie und Geodynamik, https://www.zamg.ac.at/cms/de/klima/klima-aktuell/unwetterchronik), (5) the Drought Management Center for Southeastern Europe (DMCSEE, http://www.dmcsee.org/) covering Slovenia and Slovenian text-reports, and (6) the media archive of the Swiss information platform Drought-CH (Zappa et al. (2014), https://www.drought.ch/). Most of the reported impacts of the German, Italian and Slovenian text-reports stem

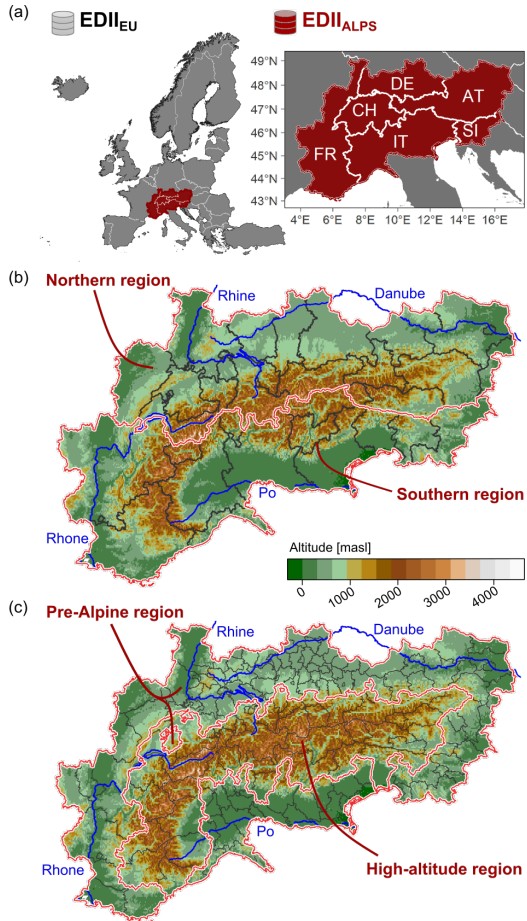

**Figure 1.** (a) The EDII_EU coverage of the whole of Europe as in Stahl et al. (2016) integrating the area covered by the newly updated Alpine Drought Impact report Inventory (EDII_ALPS). The EDII_ALPS paired domains for the analysis are: (b) the Northern and Southern region divided by grouped NUTS 2 regions, and (c) the pre-Alpine and high-altitude region divided by grouped NUTS 3 regions.

from newspaper articles, whereas the platforms Propluvia, Unwetterchronik, DMCSEE, Drought-CH are data archives that we (re-)investigated to extract and translate drought impact information to EDII fulfilling the described requirements.

Complying with the EDII guidelines (www.geo.uio.no/edc/droughtdb/img/Guidelines_EDII_Webversion.pdf), these sources offered drought impact information as negative consequences of drought as text-based reports (Italian and German reports, Unwetterchronik, DMCSEE, Drought-CH). A typical recorded entry in relation to *Agriculture and livestock farming* is described in the following example: "In some regions in lower Austria the grain harvest was less than 50 %, especially for wheat and canola. [...] The first cut of grasslands summed up to only two thirds of the normal yield [...]. Higher costs for irrigation

appeared. The federal state of lower Austria supported the farmers with 1.5 million euros for the so-called 'Feeding stuff acquisition' [...]." This report was published in August, 2003, by the Austrian centre for agricultural information. Another

description related to a drought impact on management of livestock on higher-elevated pasture that was published in October, 2018 by the Unwetterchronik reads: "Due to persisting drought some meadows in the district of Landeck could not be cut a second time [...]. There were losses between 60 to 100 %. [...] Due to fodder shortage the farmers had to buy additional hay or sell their animals. [...] In Oberland the Alpine pasture farmers brought their cattle down to the valley earlier, as there was a lack of fodder and drinking water." A typical example related to water supply is the following report published in July, 2015 by a regional Italian newspaper: "In Trento fountains were closed. At the Arco municipality drought conditions are severe with water use bans. The civil protection monitors the level of the Lago delle Piazze, where different sectoral water demands can quickly worsen the current conditions." In contrast to the other sources, Propluvia offeres mapped management strategies across France, classified by increasing warning levels dependent on the drought severity. For example, the warning level 'reinforced alert' means that in the mapped region bans on watering gardens/lawns, open spaces, golf courses, and washing the car are taking place at certain times. Further, the reduction of withdrawals for agricultural purposes less than or equal to 50 % and measures prohibiting valve operations and nautical activity are applied. This way, Propluvia provides information about negative drought impacts with specific measures for the society and economy that could be translated into the EDII database entries.

For our study region we compiled data within the Alpine Space and with available information for at least NUTS 2 regions and at least with the information on a given season or month of a year in which an impact started to occur. We then classified the impact data into 13 (out of 15) categories and to 96 subtypes related to the potentially most impacted sectors proposed by the EDII: *Agriculture and livestock farming* (1), *Forestry* (2), *Freshwater aquaculture and fisheries* (3), *Energy and industry* (4), *Waterborne transportation* (5), *Tourism and recreation* (6), *Public water supply* (7), *Water quality* (8), *Freshwater ecosystems* (9), *Terrestrial ecosystems* (10), *Air quality* (13), *Human health and public safety* (14), and *Conflicts* (15). We excluded the EDIIs categories *Soil system* (11) and *Wildfires* (12) as the link between drought, impact and management in these categories is often inconclusive and other databases, such as the Forest Fire Information system (EFFIS, 2020) are more comprehensive. The final categorisation enabled the analysis of spatial distribution, temporal trends and differences by drought and impact type. The resulting dataset of this systematic collection and classification of impact data with specific focus on the Alpine Space is a sub database of the EDII. Due to the regional focus and some describend adaptions we call this database in its first version: EDII$_{ALPS}$ V1.0 available from the FreiDok repository (doi: 10.6094/UNIFR/218623, last update 8th of January, 2021).

The distributions of all reported impact categories from the EDII$_{ALPS}$ were compared with those of the (EDII$_{EU}$) including the Alpine Space (Fig. 1). Within EDII$_{ALPS}$, we compared the pre-Alpine and high-altitude region, the Northern and Southern Alpine region, as well as the countries (respectively the part of the country within the Alpine Space), and the NUTS 2 regions (Fig. 1). Because of the counts of reports differ among these areas, we additionally focused on relative proportions ('fractions') of the 13 categories, i.e. information that is independent of the overall data availability.

For the investigation of temporal trends, we aggregated EDII$_{ALPS}$ data per year. In addition, we aim to identify potential drought events from the impact perspective by comparing these aggregated yearly impacts. For the seasonal analysis, we preprocessed impact data as follows: Within the data collection, each impact was assigned to a month or season, in which the impact started to occur. Most of the time, information about the end of an impact was unavailable. In this case we assumed that

the impact only occurred in that month or season and not beyond. When a starting season is given for an impact, we assigned the season "spring" to the months March, April, May (MAM), "summer" to June, July, August (JJA), "autumn" to September, October, November (SON), and "winter" to December, January, February (DJF) for monthly impact data.

Different drought types may lead to different impacts, with $D_{SM}$ typically related to impacts in agriculture and $D_H$ typically related to impacts within a range of several water uses, such as the water supply. In this study, we focused on the $D_{SM}$ and $D_H$ type, as $D_M$ often does not lead to impacts directly. Further, impacts on agriculture and on various water uses are among the most relevant concerning drought (Stahl et al., 2016). To analyse timing, amount and the relevance of specific impacts, we re-grouped impact data to the $D_{SM}$ or $D_H$ type using the subtype category for assignment. For instance, within the category *Energy and*

*industry* (4) subtypes are e.g. 'Reduced hydropower production' (4.1) and 'Impaired production' [...] (4.2) (Stahl et al., 2016). Regardless of the corresponding primary category, these subtypes were assigned to $D_{SM}$ and/or $D_H$. For example, 'Reduced hydropower production' (4.1) is a result of low discharge and is therefore assigned to $D_H$, whereas 'reduced productivity of annual crop cultivation' (1.1) is a result of low soil moisture and therefore assigned to $D_{SM}$. Based on expert knowledge, four different people independently identified for all 96 subtypes either the assignment to $D_{SM}$ and/or to $D_H$ (final assignment

rules are presented in Table S1). The newly grouped $D_{SM}$ and $D_H$ impacts were then analysed for seasonal occurrence in the different domains ($EDII_{EU}$, $EDII_{ALPS}$, pre-Alpine and high-altitude region, Southern and Northern region). We calculated smoothed seasonal "impact regimes" as loess curves, i.e. by local regression (Cleveland, 1979).

## 2.3    Hypothesis testing

The following hypotheses on spatial differences were tested: the distribution of impact categories of the $EDII_{ALPS}$ differed

from that of the $EDII_{EU}$, the distribution of the high-altitude region from that of the pre-Alpine region, the distribution of the Northern region from that of the Southern region. Additionally, we tested the hypothesis that the distribution of reported impact categories of $EDII_{ALPS}$ differed between the Alpine countries and the NUTS 2 regions. We tested as well, if the distribution of impact categories differed for the years between 1975 and 2020. At last, we tested whether the distribution of reported impact categories differed for the season spring, summer, autumn and winter in the domains ($EDII_{EU}$, $EDII_{ALPS}$, pre-Alpine,

high-altitude, Northern and Southern region).

For the statistical analyses, we used the compiled impact data as count data. We applied the *Pairwise Wilcoxon Rank Sum Test* to test whether the fraction of the counted data were significantly different. The test analysed if the central tendencies of the distributions between the groups differed, such as between the paired domains, the countries, the NUTS 2 regions, and the seasons (Cuzick, 1985). With the p-value $\leq 0.05$, we rejected the null hypothesis that tendencies among the tested groups had

been the same and concluded a statistically significant difference between them. In addition, if we tested more than two groups, this test allowed us to identify which group(s) was (were) significantly different, e.g. when we tested all 35 NUTS 2 regions.

## 3 Results

### 3.1 Spatial differences across domains and countries

With our update at the time of this study EDII$_{EU}$ for the whole of Europe contains in total more than 10,100 reported drought impacts for NUTS 2 and 3 regions. For the Alpine Space, our newly compiled impact data has raised substantially the number of archived reports from 1412 (i.e. status of EDII from September 2019) to more than 3,000 (EDII$_{ALPS}$ V1.0, last update 8th of January, 2021). For the EDII$_{ALPS}$ we could add missing drought impact reports especially located in Austria, Slovenia, Italy and France (for detailed information on the numbers and distribution of our update refer to the Supplementary material, Sect. 2). We observed substantial differences between the amounts of reported impacts across the domain of the Alpine Space (Fig. 2) with more reports located in the Northern than in the Southern region and more in the pre-Alpine than in the high-altitude region. The *Pairwise Wilcoxon Rank Sum Test* depicted significant differences between the impact category distributions of the EDII$_{ALPS}$ and the EDII$_{EU}$, but not between the other pairs (Table 1). Among the Alpine countries, the test identified Slovenia to be significantly different from Austria and Germany. At the smaller scale, several NUTS 2 regions located in Italy differed to NUTS 2 regions in Austria, Switzerland and Germany (Table 1, Fig. 2a).

In addition to the test results, relative fractions provide further information (Fig. 2). The EDII$_{ALPS}$ and the EDII$_{EU}$ report the same two most frequent impact categories: *Agriculture and livestock farming* (EDII$_{ALPS}$: 48 %, EDII$_{EU}$: 30 %) and *Public water supply* (EDII$_{ALPS}$: 19 %, EDII$_{EU}$: 24 %). More than half of all reports on drought impacts belong to these two categories in both regions. With 67 % this dominance is even stronger in the EDII$_{ALPS}$ (EDII$_{EU}$: 54 %). The fraction of reported impacts in *Agriculture and livestock farming* is clearly higher in all domains in the EDII$_{ALPS}$ compared to the fraction of the EDII$_{EU}$. Especially, the Southern region presents almost 60 % of the reported impacts in this category, with more than half of all reports related to the subtypes 'Reduced productivity of annual [...]' (1.1) or '[...] permanent crop cultivation' (1.2) and 'Agricultural yield losses $\geq$ 30 % of normal production' (1.3). This is related to 96 % of all impacts in *Agriculture and livestock farming* on country-level in Slovenia. The subtype 'Reduced availability of irrigation water' (1.4) is the most prominent in the high-altitude region. Impacts related to the second most important category *Public water supply* are less frequent in the EDII$_{ALPS}$ compared to the EDII$_{EU}$. In the EDII$_{ALPS}$, the high-altitude region depicts the highest fraction with 30 %. On country-level France (Fig. 2a) stands out with 36 %. Both regions exceed the overall fraction on this category in the EDII$_{EU}$. The most common subtype in this category is 'Bans on domestic and public water use' (7.3).

The third most frequent category in the EDII$_{EU}$ is clearly identified with *Freshwater ecosystems* (11 %). For the EDII$_{ALPS}$' third rank, we identify *Forestry* (7 %) with a slightly higher fraction than *Freshwater ecosystems* (6 %). 16 % of all entries in the German part of the Alpine countries and 13 % of the Swiss entries relate to *Freshwater ecosystems*, and thus, exceed the fraction of the EDII$_{EU}$. Most of these impacts are located at the Rhine river and most frequent with the subtype 'Increased mortality of aquatic species' (9.1). The EDII$_{ALPS}$' impacts on *Forestry* (7 %) are as well mostly located in the German (19 %) and Swiss (10 %) part of the Alpine Space (Fig. 2a). In the EDII$_{EU}$ the category *Forestry* is ranked 6th with a fraction of 6 %. In both the EDII$_{EU}$ and the EDII$_{ALPS}$, we identify the same most relevant subtype 'Reduced tree growth and vitality' (2.1).

The EDII$_{ALPS}$ presents these four described categories *Agriculture and livestock farming*, *Public water supply*, *Freshwater ecosystems* and *Forestry* among the most important ones for all domains, but as well on the country-level with the following differences (Fig. 2a,b). As already mentioned, reports located in Slovenia are clearly dominated by the category *Agriculture and livestock farming* (96 %). More than 70 % of the impacts located in France (70 %), Italy (78 %), and Austria (72 %) report drought affecting *Agriculture and livestock farming* and *Public water supply* with switching relevance. Impacts located 230 in Germany and Switzerland present less dominance by these two categories, as *Forestry* and *Freshwater ecosystems* play as well a major role.

Regarding the other categories and subtypes, we observe smaller differences. Impacts related to *Waterborne transportation* present the 4th and 5th highest fraction (EDII$_{EU}$: 10 %, EDII$_{ALPS}$ 5 %). The impact frequency is lower in the EDII$_{ALPS}$, but high in the French part with 18 % and in the high-altitude region with 8 % of all impacts related to this category (Fig. 2a,b). 235 Whereas the EDII$_{EU}$ relates most to the subtype 'Impaired navigability of streams (reduction of load [...])' (5.1), the majority of the impacts in the EDII$_{ALPS}$ are not tangible to a specific subtype. They relate to 'Others' (5.3), with a majority from the French database Propluvia mapping 'measures prohibiting valve operation, nautical activity', which we could not clearly assign to any subtype of the category *Waterborne transportation*. In the EDII$_{EU}$ and the EDII$_{ALPS}$, the category *Water quality* presents a fraction of 7 % and 4 %, both with the most frequent subtype '(Temporary) Water quality deterioration / problems 240 of surface waters [...]' (8.2). Thus, the frequency of impacts reported for this category is lower in the EDII$_{ALPS}$. Most of these impactcs are located in Italy with 7 % meeting the reporting frequency of the EDII$_{EU}$ (Fig. 2a,b). The categories *Air quality*, *Freshwater aquaculture and fisheries*, *Tourism and recreation*, *Terrestrial ecosystems*, *Energy and industry*, *Human health and public safety* and *Conflicts* do not depict an obvious relevance in any of the analysed regions.

## 3.2 Temporal trends and drought years

Before the year 1950 the EDII$_{ALPS}$ only contains a small number of reported impacts (n: 342), covering eight categories, dominated by *Agriculture and livestock farming* (n: 270) and followed by *Public water supply* (n: 23), *Energy and industry* (n: 16), *Forestry* (n: 13) and *Human health and public safety* (n: 11). In this early time-period most impacts are reported in Switzerland, Germany, Austria and France. The number of reported impacts increases substantially after 1975, and again after 2000 and 2010 (Fig. 3a,b). 2019 and 2020 have less impact reports. After 1975, the years 1976 (n: 120), 2003 (n: 401), 2015 (n: 250 452), and 2018 (n: 364) show substantially more impacts than all other years. The *Pairwise Wilcoxon Rank Sum Test* confirms the years 2003, 2015 and 2018 to be significantly different from others. Thus, our impact data represents four specific drought years. Beside the increase of collected impacts over time, the diversification increase as well.

Comparing the identified drought events (Fig. 3c), the category *Agriculture and livestock farming* is extremely dominant in 1976 (82 %) and much less present in the years 2003, 2015 and 2018 (33 %, 28 %, and 27 % respectively). In 1976, the 255 second most reported impacts are on *Forestry* (12 %), whereas the other categories are not reported or with low relevance. The year 2003 shows high fractions for *Public water supply* (24 %) followed by *Freshwater ecosystems* (20 %) and *Water quality* (12 %), but still dominated by *Agriculture and livestock farming* (33 %). The most frequent subtypes during this drought event are 'Agricultural yield losses $\geq$ 30 % of normal production [...]' (1.3) and 'Increased mortality of aquatic species' (9.1).

**Table 1.** Results of the *Pairwise Wilcoxon Rank Sum Test* comparing the central tendencies between different regions: the paired domains and smaller subregions. For subregions results are shown only for combinations differing by a p-value $\leq 0.15$.

| Regions | P-value[1] |
|---|---|
| Alpine Space vs. Europe | 0.022* |
| Pre-Alpine region vs. high-altitude region | 0.077 |
| Northern region vs. Southern region | 0.097 |
| Alpine countries (n = 5) | |
| *Austria:Slovenia* | 0.010* |
| *Switzerland:Slovenia* | 0.063 |
| *Germany:Slovenia* | 0.035* |
| Alpine NUTS 2 regions (n = 35) | |
| *Provincia Autonoma di Bolzano/Bozen:Niederösterreich* | 0.115 |
| *Provincia Autonoma di Bolzano/Bozen:Ostschweiz* | 0.143 |
| *Provincia Autonoma di Trento:Ostschweiz* | 0.134 |
| *Provincia Autonoma di Bolzano/Bozen:Freiburg* | 0.031* |
| *Provincia Autonoma di Trento:Freiburg* | 0.050* |

[1] '*' significant at p-value $\leq 0.05$.

The fractions of 2015 differ to the one of 2003 as follows. In 2015, we observe more reports related to *Forestry* (8 %) and *Waterborne transportation* (6 %), but less to *Agriculture and livestock farming* (28 %), *Freshwater ecosystems* (13 %), *Water quality* (8 %) and *Energy and industry* (3 %). Further, the categories *Terrestrial ecosystems*, *Human health and public safety* and *Tourism and recreation* are represented with low relevance. In addition, no subtype presents the fraction $\geq 10$ % (Fig. 3c). The fractions of the year 2018 are comparable. Impacts related to *Agriculture and livestock farming*, *Public water supply* and *Freshwater ecosystems* do not change their fractions remarkably between 2003 and 2015. Within these categories, the subtypes 'Reduced availability of irrigation water' (1.4) and 'Bans on domestic and public water use' (7.3) are more frequent in 2018 compared to 2015 and 2003. Furthermore, the year 2018 shows higher fractions in *Forestry* (14 %), *Waterborne transportation* (10 %) and *Tourism and recreation* (6 %) and less in *Energy and industry* (3 %) and *Water quality* (1 %).

### 3.3 Seasonal patterns

Across all domains reported drought impacts occur mostly in summer, followed by autumn, spring and fewest in winter (Fig. 4). Significant seasonal differences are found in the $EDII_{EU}$ and in the pre-Alpine and high-altitude region of the $EDII_{ALPS}$ (Table 2) with the summer always significantly different from the winter. We identify low p-values for the summer differing from spring, but not between summer and autumn. Reported impacts during autumn present no significant differences compared to the other seasons.

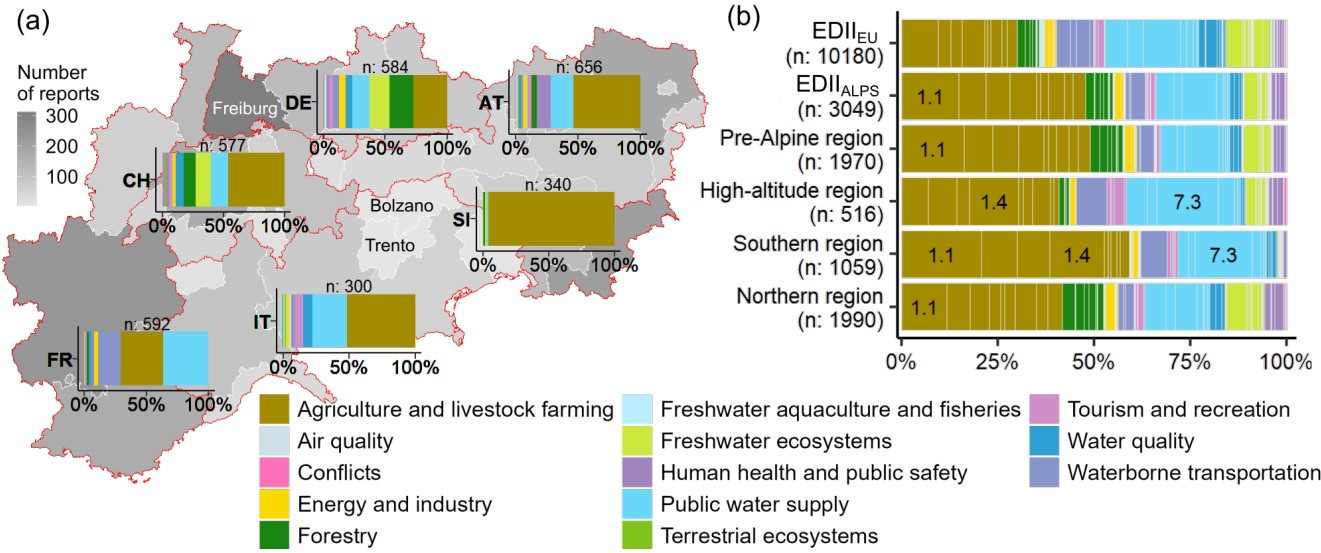

**Figure 2.** Reported impacts in 13 coloured categories by region (n = total no. of reports per region). (a) Fraction of impact category per country. NUTS 2 regions with significant differences are labeled (see Table 1). (b) Fraction of impact categories covering several subtypes (faint lines) for the domains. Subtypes with a fraction ≥ 10 % per region are labeled.

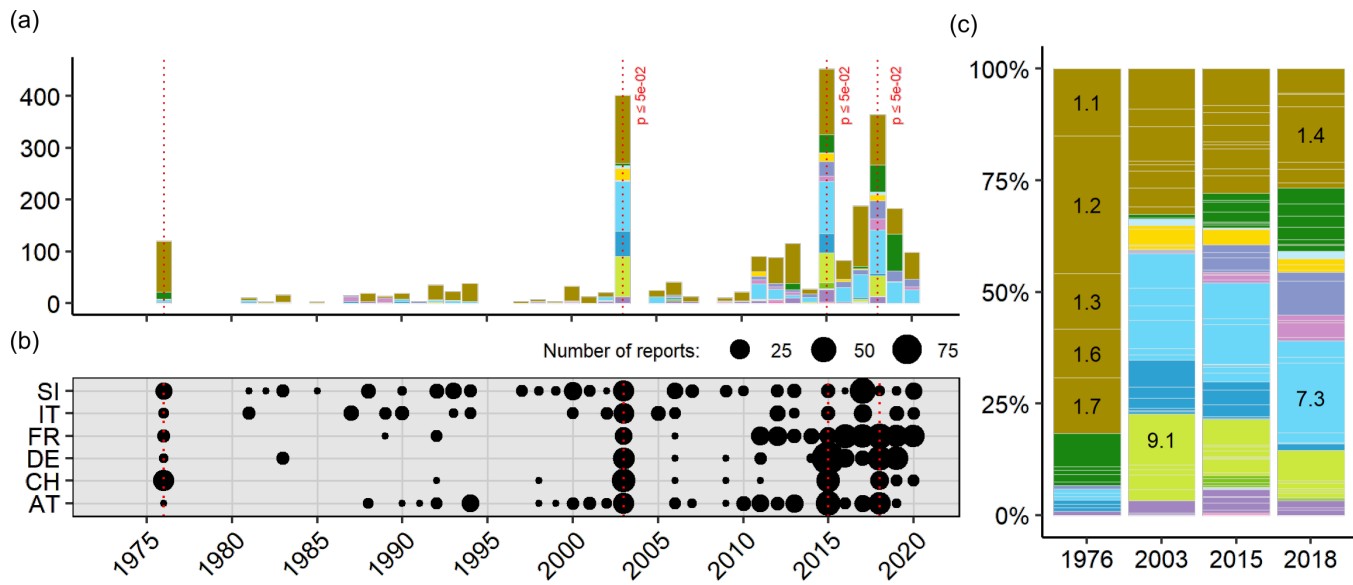

**Figure 3.** Reported impact categories between 1975 and 2020. (a) Number of reports per year and identified drought events (red dotted line) with significantly different years labeled. (b) Number of all reports per country and year. (c) Fraction of impact categories (coloured) and their subtypes (faint lines) for drought events. Subtypes with a fraction ≥ 10 % per region are labeled. Colours see Fig. 2.

Although the seasonality of EDII$_{ALPS}$ and the EDII$_{EU}$ is rather similar for the domains, some categories showed differences. Impacts related to *Agriculture and livestock farming* occur in all seasons in the EDII$_{ALPS}$, wherefore it is the dominating category throughout the year excluding winter. In the EDII$_{EU}$ this category dominates only during summer and *Public water supply* is the most dominant category at other times of the year (Fig. 4a,b). Regarding *Agriculture and livestock farming*, we observe most of these impacts in the summer in all domains covered by the EDII$_{ALPS}$, with a strong increase before summer, followed by a strong decrease after summer. The pre-Alpine and Southern region show a clear increase of impacts in March and April (Fig.4c,f), and the high-altitude region has substantially higher counts in September and October (Fig. 4d). In the EDII$_{ALPS}$, the spring and summer impacts in this category relate most to the subtype 'Reduced productivity of [...] crop cultivation' (1.1, 1.2), whereas the impacts in autumn relate most to 'Reduced availability of irrigation water' (1.4). The EDII$_{ALPS}$ reported impacts in *Public water supply* are less frequent compared to the ones by the EDII$_{EU}$, and especially in the first months of the year till May. The high-altitude region shows the most different pattern compared to the other domains by the EDII$_{ALPS}$ (Fig. 4a,c,d,e,f). In these domains, the monthly sums of this category do not increase in spring, but start to accumulate from May to August with high fractions in the high-altitude region. There, impacts on *Punblic water supply* are less but still frequently reported in autumn. Especially in November and December, the reported impacts show the same fractions as those related to *Agriculture and livestock farming*. In the EDII$_{ALPS}$ the most frequent subtype 'Bans on domestic and public water use' (7.3) dominates this category, whereas in the EDII$_{EU}$, the subtypes 'Local [...] and region-wide water supply shortage / problems' (7.1, 7.2) are as well prominent.

The EDII$_{ALPS}$' category *Freshwater ecosystems* reports most impacts in summer and autumn and almost none in spring and winter, with reports mostly located in the Northern region respectively pre-Alpine region (Fig. 4c,e). The EDII$_{EU}$ presents as well most counted reports in the summer months, but the fractions of this category spread more equally across the seasons. Further, we identify most impacts related to *Forestry* to occur in summer for both, in the EDII$_{ALPS}$ and in the EDII$_{EU}$. Within the domains of the EDII$_{ALPS}$, we observe the seasonal fraction of *Forestry* impacts to be varying in relative terms. We depict the lowest relative fractions in autumn, which is due to higher counts in spring and summer especially in the Northern region (Fig. 4e) and higher counts in winter in the high-altitude region (Fig. 4d). In the EDII$_{ALPS}$ and in the EDII$_{EU}$, we find most frequent reports in *Waterborne transportation* from high summer to early September and with highest seasonal fraction in autumn (Fig. 2a,b). Impacts related to *Tourism and recreation* differ in their seasonal fractions between the domains. In the EDII$_{ALPS}$, these impacts are mostly winter impacts with a majority located in the high-altitude and Southern region (Fig. 4a,d,f). Additionally, the high-altitude region shows higher fractions in spring for this category. Though the pre-Alpine and Northern region report most impacts in summer, similar to EDII$_{EU}$, these records are few compared to the more frequent categories, such as *Agriculture and livestock farming* and *Public water supply*. We also find a few impacts on *Air quality* in the EDII$_{ALPS}$ and the EDII$_{EU}$. In the EDII$_{ALPS}$, this category together with *Tourism and recreation* are the only ones reported most in autumn (*Air quality*) and winter (*Tourism and recreation*).

**Table 2.** Results by the *Pairwise Wilcoxon Rank Sum Test* comparing the central tendencies between the different seasons in all domains. Only combinations differing by a p-value ≤ 0.15 are shown.

| Regions | Seasons | P-value[1] |
|---|---|---|
| Europe | Summer:Winter | 0.030* |
| Alpine Space | Summer:Spring | 0.117 |
| | Summer:Winter | 0.053 |
| Pre-Alpine region | Summer:Spring | 0.109 |
| | Summer:Winter | 0.019* |
| High-altitude region | - | - |
| Northern region | Summer:Spring | 0.067 |
| | Summer:Winter | 0.017* |
| Southern region | - | - |

[1] '*' at significant p-value ≤ 0.05. '-' no combination differing with p-value ≤ 0.15.

## 3.4   Impacts related to drought types

We re-grouped 42 out of 96 subtypes by their related drought type $D_{SM}$ and $D_H$ (Fig. 5a). Twelve subtypes are classified as $D_{SM}$ impacts with most of them from the categories *Forestry* and *Agriculture and livestock farming*. Further, we classify 32 subtypes as $D_H$ impacts with a majority from the categories *Water quality*, *Public water supply*, and *Freshwater ecosystems*.

Two subtypes are classified into both, the $D_{SM}$ group and the $D_H$ group: 'Lack of feed/water for livestock' (1.7) and 'Lack of feed/water for wildlife' (10.8), as low soil moisture as well as low discharge and/or groundwater storage are appropriate drivers. For 54 subtypes neither $D_{SM}$ nor $D_H$ is considered as a clear trigger. They are classified into a group of indirect impacts that include impacts less tangible to specific drought conditions and strongly dependent on market situations, management and governance aspects. We find most of these ambiguous subtypes in the categories *Energy and industry*, *Human health and public safety*, *Air quality*, *Conflicts*, *Freshwater ecosystems* and the subtype 'Increased costs/economic losses' in several categories (all assignments are presented in Table S1).

With this classification scheme, the fraction of $D_{SM}$ and $D_H$ impacts differs clearly per domain (Fig. 5). In the $EDII_{ALPS}$ 38 % of all impacts are assigned to $D_{SM}$ and 40 % to $D_H$. In the $EDII_{EU}$, $D_{SM}$ impacts are less (23 %) and $D_H$ impacts (55 %) more frequent (Fig. 5a,b). In the pre-Alpine region, 40 % of the impacts are assigned to $D_{SM}$ and 39 % to $D_H$. In the high-altitude region, we assign less impacts to $D_{SM}$ (25 %) and more to $D_H$ (52 %) (Fig. 5c,d). The Northern and Southern region compared differ less with 36 % and 41 % assigned to $D_{SM}$ and 39 % and 42 % assigned to $D_H$ (Fig. 5e,f). The Southern region shows the greatest fraction of combined $D_{SM}$ and $D_H$ impacts (83 %) among all domains.

In the $D_H$ impact group, the most frequent subtypes are 'Bans on domestic water use' (7.3), 'Local water supply shortage/problems' (7.1), 'Reduced availability of irrigation water' (1.4), 'Regional shortage of feed/water for livestock' (1.7), and

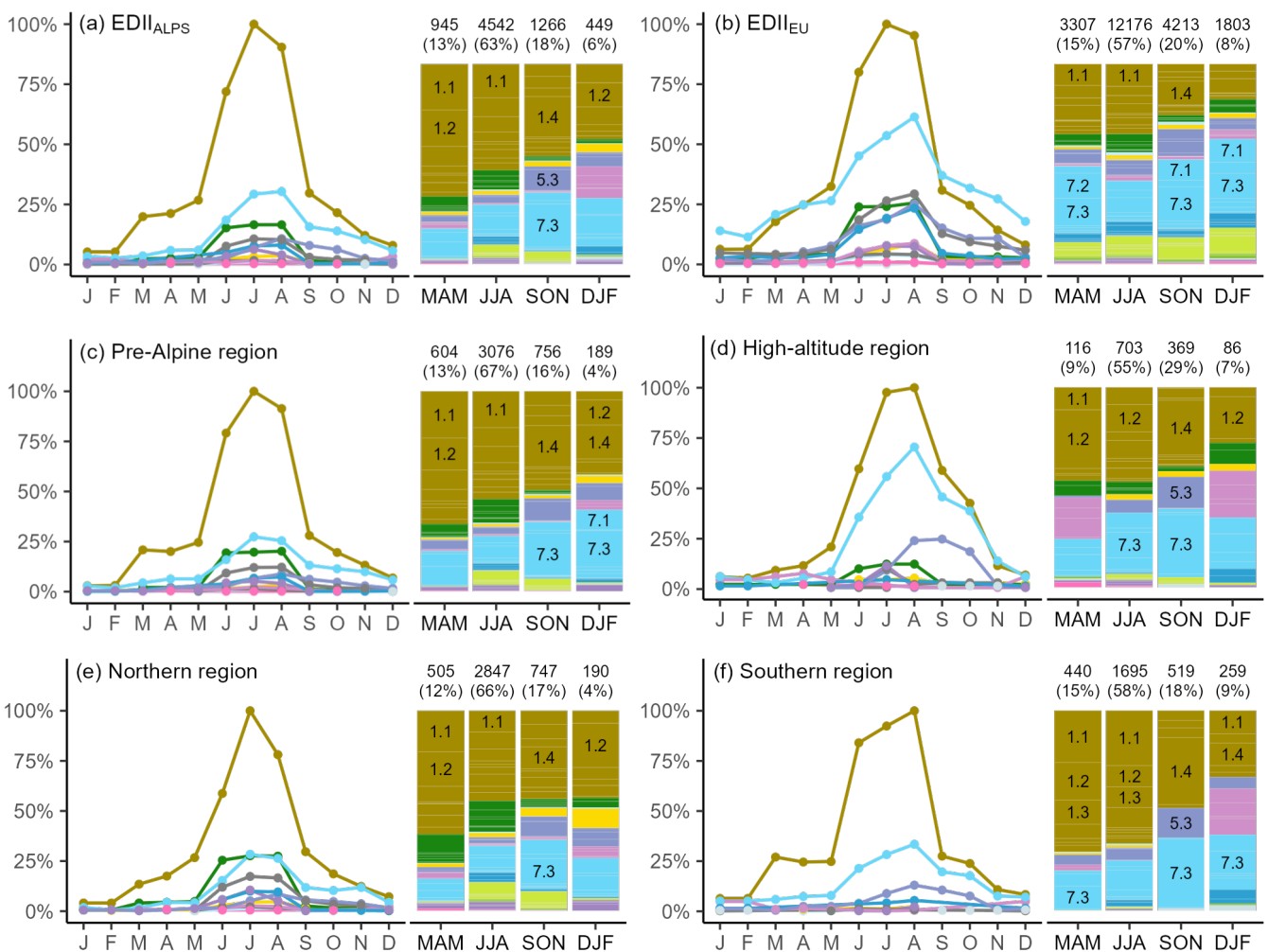

**Figure 4.** Annual distribution of reported impact categories per month (line diagrams) and season (bar plots) for the (a) EDII$_{ALPS}$, (b) EDII$_{EU}$, (c) pre-Alpine region, (d) high-altitude region, (e) Northern region, (f) Southern region. Monthly percentages relate to the sum of all impacts per month and category with 100 % corresponding to the month with most impacts. Total counts of each season are given on top of the bars, the fraction in brackets relates to the amount of impacts assigned to the season. Subtypes with a fraction ≥ 10 % per season are labeled. Colours see Fig. 2.

'Increased mortality of aquatic species' (9.1). We find the largest proportional differences between the pre-Alpine and high-altitude region among the subtypes 'Reduced availability of irrigation water' (1.4), and 'Bans on domestic and public water use' (7.3). Accordingly, we observe a shift among these subtypes between the Northern and Southern region, but additionally for the subtype 'Increased mortality of aquatic species' (9.1). Both differences confirm the previous results. In the D$_{SM}$ impact group, the most frequent reports are in the category *Agriculture and livestock farming* and about the subtypes 'Reduced

productivity of annual crop cultivation' (1.1), 'Reduced productivity of permanent crop cultivation '(1.2), 'Agricultural yield losses ≥ 30 %' (1.3) and 'Regional shortage of feed/water for livestock' (1.7). The different fractions for these subtypes depend on the domain. The Southern region reports the largest fraction of impacts about 'Reduced productivity of annual crop cultivation' (1.1), 'Reduced productivity of permanent crop cultivation' (1.2), 'Agricultural yield losses ≥ 30' (1.3). Within the $D_{SM}$ impacts, *Forestry* impacts are substantially less frequent and non-existent in the Southern region.

The annual regime curves of the $D_{SM}$ and $D_H$ impacts are based on monthly reported impact sums. For all domains most $D_{SM}$ and $D_H$ impacts occur in summer and early autumn as already shown in the previous results. In case of $D_{SM}$ impacts, the high occurrence peaks are driven by 'Reduced productivity of annual crop cultivation' (1.1). Regarding the $D_H$ impacts, the high peaks correspond to 'Local water supply shortage / problems' (7.1) in the Northern region and to 'Bans on domestic and public water use' (7.3) in all other domains.

According to the total counts of the grouped impacts by drought types, the $EDII_{ALPS}$ has a higher $D_{SM}$ impact peak occurring earlier in the year (June-July) than the $D_H$ impact peak (July-August) (Fig. 5a). In addition, the increase and decrease of the $D_H$ curve happens later in the year than that of the $D_{SM}$ curve. Thus, between March and July $D_{SM}$ impacts show higher fractions, while between September and December $D_H$ impacts show higher fractions. The $EDII_{EU}$ contrasts these results (Fig. 5b), as the records across Europe depicts the highest peak of $D_H$ impacts in the same summer months compared to the $D_{SM}$ impact peak. Within the $EDII_{ALPS}$ the delayed $D_H$ peak and the higher fractions of $D_H$ impacts between September and December is confirmed by all paired domains (Fig. 5c,d,e,f). The most different pattern is found for the high-altitude region with the latest onset of the $D_H$ impact curve and a delayed peak between August and September (Fig. 5d). Furthermore, this is the only domain within the Alpine Space with a higher $D_H$ impact peak, and subsequently, the highest fractions between July and December.

## 4   Discussion

### 4.1   Drought impacts across the European Alpine region

Although the Alpine Convention has started to raise the topic, drought impacts in different regions of the Alpine and pre-Alpine area have not yet been formally compared. Assembling the $EDII_{ALPS}$, an inventory of drought impact reports, enables a first overview and some regional comparisons. Any collection of drought impact data is a challenging task due to the lack of a clear definition of drought impact's onset and end (Bachmair et al., 2015). The report collection enables this link, as we only compiled reports clearly stating drought to be the cause of the described impact. Some impacts can be measured and are therefore easier to collect and hence more consistent through time and space (e.g., the agriculture yield losses), but most of them are hard to quantify (Logar and van den Bergh, 2013). For most of the impacts no continuous data is available, for which the text-reports proved to be a suitable surrogate and are worthwhile to collect (Bachmair et al., 2016; Hayes et al., 2012). Nonetheless, not all impacts are reported or only reported locally, in which case the compiled information in $EDII_{ALPS}$ V1.0 may still have gaps. 70 % of all reports stem from the Northern region. We have been expecting the Southern region characterised by Meditarranean climate to be at least equally impacted but assume our impact data might be regionally more complete in the Northern region. On the other hand Mediterranean regions may be better prepared for drought or as dry and hot

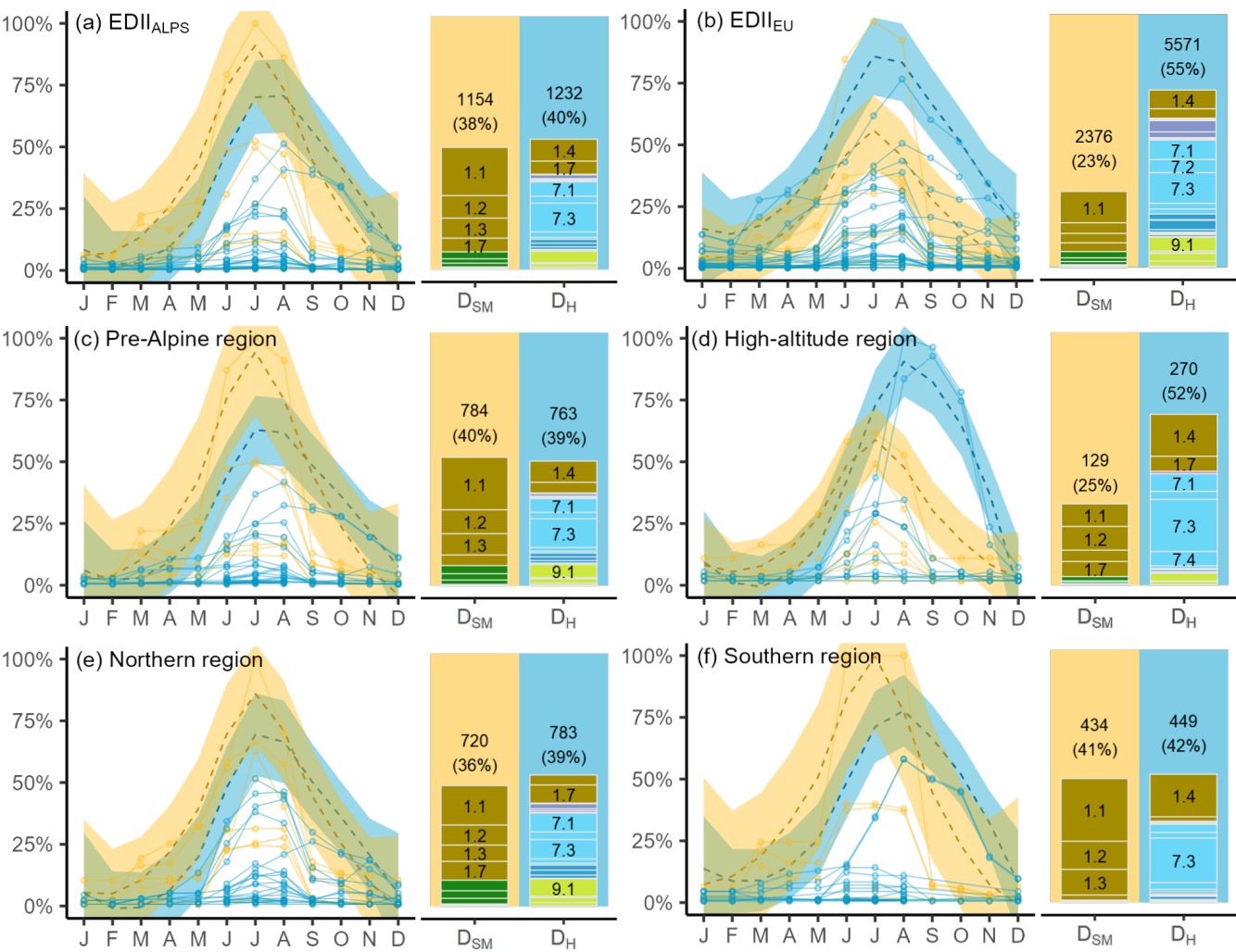

**Figure 5.** Impact subtypes assigned to $D_{SM}$ (yellow) or $D_H$ (blue) per month (line diagram) and drought type (bar plot) for the (a) EDII$_{ALPS}$, (b) EDII$_{EU}$, (c) pre-Alpine region, (d) high-altitude region, (e) Northern region, (f) Southern region. Seasonal regimes for $D_{SM}$ (yellow lines) and $D_H$ impacts (blue lines) are loess curves with standard errors (dotted line with coloured shape). Monthly percentages (solid lines) relate to the sum of all impacts per month and subtype with 100 % corresponding to the month with most impacts. Total counts of $D_{SM}$ and $D_H$ impacts are given on top of the bars, the fraction in brackets relates to the amount of impacts assigned to the season. Subtypes with a fraction $\geq$ 5 % are labeled. Colours see Fig. 2.

conditions are more common are less concerned with reporting them. In addition, we might have missed drought impacts that are not reported to be associated with drought due to the typical perception that mountain regions are water-rich. Especially winter drought might be still underestimated, as impacts caused by low snow accumulation can occur delayed and therefore might not be associated with climatic conditions months ago. Despite these barriers, the amount of impact data we collected for

the EDII$_{ALPS}$ (n: 3243) demonstrates that drought impacts have been present in the European Alpine region. Despite the humid mountain climate, 30 % of all impact data across Europe is located in our study region. Subsequently, this study confirms the relevance to understand the role of drought as well in the European Alpine region and not only in low-altitude areas.

Most of the EDII$_{ALPS}$' recent data stem from newspaper articles, web pages, governmental reports and scientific articles and specialty databases. We analysed the impact data differences between the EDII$_{ALPS}$ and the EDII$_{EU}$, as well as between the paired domains (pre-Alpine and high-Alpine, Northern and Southern region) and between political units (countries, NUTS 2 regions) within the Alpine Space.

Similar to all of EDII$_{EU}$, most reported impacts in the EDII$_{ALPS}$ fell into the category *Agriculture and livestock farming*, 375  specifically to the subtype 'Reduced productivity of annual crop cultivation' (1.1). In contrast to the Northern region, the Southern region is known for less precipitation and higher air temperatures due to the Mediterranean climate. Even though the region adapted partly to the climatic conditions with permanent irrigation systems (Yves et al., 2020), we found substantially more impacts related to *Agriculture and livestock farming*. Haslinger and Blöschl (2017) showed that drought frequencies were higher in the Southern region and hence also the exposure for impacts. Relatively, even more impacts in *Agriculture and live-* 380  *stock farming* were reported by the EDII$_{ALPS}$ than by the EDII$_{EU}$. It should be kept in mind that the EDII$_{ALPS}$ covers the 'Alpine Space', and therefore includes not only the mountains but large foothill and foreland areas as well (Fig. 1). Besides urban areas such as Vienna, Milan or Zurich, the Alpine Space is known for its agricultural crop production in lower elevation areas and for mountain pastures for beef and dairy production typical in higher elevated areas (Jäger et al., 2020). Subsequently, in this region *Agriculture and livestock farming* is among the most relevant economic sectors (Flury et al., 2013). This dependence 385  potentially affects the vulnerability and hence has driven the number of reported impacts we compiled compared to the EDII$_{EU}$ with less economic dependency on the agricultural sector.

The second most frequent impact category in the EDII$_{ALPS}$ related to *Public water supply*. Compared to the EDII$_{EU}$ this impact category was less relevant, hence confirming the typical association that the European Alpine region is a water-rich Water Tower (Viviroli et al., 2007). Nonetheless, the amount of reports for the subtype 'Bans on domestic and public water 390  use' (7.3) emphasizes the need to develop management strategies. Specifically, for periods of particular high water demand compared to its actual availability, e.g. periods with high demand for supplementary irrigation or for high touristic uses. The fraction of impacts related to *Public water supply* was highest in the high-altitude region, but it was not high in absolute number. To fully explain this result, further studies need to investigate whether upstream headwater areas have to deal with impacted water supply, or less access to stored water compared to lower (downstream) areas with allocated water.

Impacts classified into the category *Freshwater ecosystems* were less reported and it may be concluded that they might be less relevant in the EDII$_{ALPS}$ compared to the EDII$_{EU}$. However, in the Northern region the categories fraction was higher, mainly driven by reports from Switzerland and Germany exceeding the fraction of the EDII$_{EU}$. Most of these reports were located along the river Rhine and associated with high water temperatures and less oxygen saturation in the year 2003. Drought impacts on *Forestry* were more relevant in the EDII$_{ALPS}$. This category ranked 3rd compared to 6th in the EDII$_{EU}$. Most of these impacts 400  were located in Germany and Switzerland. The majority of German reports originated from governmental reports by national and regional agencies, such as the forest institutes by the federal state Baden-Wurttemberg or Bavaria that regularly publish

articles about the forest conditions in Southern Germany. We found similar reports in Switzerland, but not in the other Alpine countries. Thus, EDII$_{\text{ALPS}}$ likely still misses *Forestry* impacts in other regions. The drought impact categories *Waterborne transportation* and *Water quality* appear to be less relevant in the EDII$_{\text{ALPS}}$ compared to the EDII$_{\text{EU}}$. Reasons may include *Waterborne transportation* to take place in the lowlands and *Water quality* benefitting from less pollution and environmental degradation. Other impact categories did not show substantial differences. Even though hydropower has a greater importance in mountainous regions, the number of reported impacts in *Energy and industry* in EDII$_{\text{ALPS}}$ did not differ compared to EDII$_{\text{EU}}$. One reason among others might be related to the hydropower generated from reservoir storage, which makes operations more resilient towards drought events.

The differences between the impact data of the Alpine countries are influenced by the sources we investigated. Comparable to EDII, the Unwetterchronik, Drought-CH as well as the text-report collection for Germany and Italy offer varied information about the droughts' impacts. In contrast to these more broad databases, the bulletins from DMCSEE and the Slovenian text-reports are more focused on impacts related to agriculture and the French "Propluvia" informs on water restrictions in drought periods. The resulting report based impact data collection EDII$_{\text{ALPS}}$ is therefore influenced by national priorities and different collection efforts effects, a limitation Stahl et al. (2016) as well discussed. Our statistical tests confirmed this spatial heterogeneity. Nonetheless, we should consider the national focus as valued information, because this likely depicts the current major challenges of drought management on the national scale, but further efforts might focus on complementing them. Despite these differences, EDII$_{\text{ALPS}}$ depicts plausible patterns in altitudinal and climatic subregions in the Alpine Space.

## 4.2 Drought events and temporal impact trends

The reported impacts compiled in EDII$_{\text{ALPS}}$ identified four specific drought years in the Alpine Space: 1976, 2003, 2015, and 2018. Studies focusing on drought events across the European mountain region are scarce, wherefore it is difficult to relate the impact based drought years to climatologically identified drought events in the Alpine Space. Haslinger and Blöschl (2017) presented no clear drought trend from 1801 to 2010, but ranked the 2003 drought as the second most severe event, likely resulting in our substantial increase of reported impacts. Brunner and Tallaksen (2019) compared the drought events 2003, 2015 and 2018 due to the severe ecological, economic and social impacts in Switzerland confirming our identified drought years with the EDII$_{\text{ALPS}}$. On the European scale several studies call the years 1976, 2003, 2015 and 2018 'drought events' identified by various selection methods, most based on drought indices respectively the detection of anomalies (Parry et al., 2012; Spinoni et al., 2015; Laaha et al., 2017; Hoy et al., 2020; Hari et al., 2020). This leads to the assumption that the EDII$_{\text{ALPS}}$ complements these studies with evidence of impacts (Fig. 3a) reflecting the hazard potential of severe drought events. Less severe droughts or droughts with less public attention might not be sufficiently represented.

Further, the EDII$_{\text{ALPS}}$ suggests a diversification of impacts over time, confirming observations in earlier studies with the EDII$_{\text{EU}}$ (Stahl et al., 2016). The impacts of the drought in 1976 differed substantially from later events, as more than 80 % of all impacts related to *Agriculture and livestock farming*, followed by Forestry. Parry et al. (2012) explained the unprecedented severity of the drought event 1975-76 by the consecutive occurrence of winter and summer drought likely explaining the high number of impacts archived in EDII$_{\text{ALPS}}$. The dry starting conditions in spring and early summer not only affected *Agriculture*

*and livestock farming*. In addition, possibly *Forestry* was as well affected already during winter, but showed the impacts delayed the next summer. An effect that the data also suggest for the droughts in 2015 and 2018. In terms of higher impact diversification the years 2003, 2015 and 2018 depicted a more comparable picture than 1976. 2003 differed the most from 2015 and 2018 with the high fractions of the category *Freshwater ecosystems* and *Water quality*. According to the previous section, these impacts

correspond to the river Rhine in Northern Switzerland and Southwest Germany. The characteristics of the 2003 drought have been a combination of heatwaves especially in central Europe combined with very dry summer conditions Schär and Jendritzky (2004) leading to high water temperatures and low oxygen levels firstly reported as impacts in the category *Water quality*. Finally, these impacts have led to the great fish dieback, reflected by the subtype 'Increased mortality of aquatic species' (9.1) of the category *Freshwater ecosystems*. Blauhut et al. (2015) associated the report's increase with the new EU Water

Framework Directive, which raised the topic's attention. The years 2015 and 2018 showed substantially increasing reports related to *Forestry*. Laaha et al. (2017) compared the drought 2015 to 2003 and presented 2015 with drier starting conditions along the Northern Alps and the notably longer duration in Southern Germany. Hari et al. (2020) characterised the drought 2018 persisting into 2019, unlike 2003, where ecosystem carbon and energy fluxes recovered early after the summer. Thus both droughts in 2015 and 2018 were characterised by a longer duration that accumulates in dry soils and subsequently $D_{SM}$. The

increased fractions of *Forestry* impacts in 2015 and 2008 compared to 2003 likely reflect these persistent characteristics of $D_{SM}$ accumulating longer and thus depict delayed impacts. Trotsiuk et al. (2020) showed substantially stronger negative trends by the species *Fagus sylvatica* and *Picea abies* of forest productivity in Switzerland for the year 2018 compared to 2003. Although they did not analyse the year 2015, the results for 2018 likely correspond to accumulated effects. Ogle et al. (2000) predicted higher tree mortality following severe droughts and McDowell et al. (2010) suggested the drought-induced lower, but more

continuous mortality of tree species occurring delayed due to several interdependent physiological mechanisms. We observed a similar but less prominent pattern for the category *Waterborne transportation* between the years 2003 and 2018 that might correspond to a longer lasting $D_H$, with less discharge and/or groundwater storage over the years.

Over the whole time period, the number of collected drought impacts increased, especially after 2000 and 2010. This trend is influenced by (1) general reporting behaviour changes with digitization and online media availability (regardless of the

sources, such as scientific or newspaper articles or governmental reports), (2) accessibility to drought reports in the recent past being easier than to historic information (3) awareness of the drought hazard having increased along with climate change. For the most recent droughts, reports are yet to be published. Thus, the decreasing number of reports especially for the last two years is likely a delayed effect of publishing and collecting such text-based impact information. Nevertheless, we presented significantly different values for the years 2003, 2015 and 2018, which correspond well to the major drought events after 2000.

Additionally, 1976 depicted substantially more impact data, in a time where digitalization and the accessibility to online articles had been very different compared to the last 20 years. Thus, we expect our time trend to be biased. However, the $EDII_{ALPS}$ as a whole still depicts the major events and the tendency of increased impacts.

## 4.3 Seasonal patterns and delayed impacts

Summer and early autumn were the seasons with most drought impacts in all domains regardless of impact category or drought type. This confirms the expectation that drought impacts occur most frequently in summer. Additionally stressed by evapotranspiration, this season has the highest water demands, and hence, higher potential water shortages occur despite a mostly balanced annual precipitation in the Northern and Western parts of the Alpine Space (Kruse et al., 2010). In early autumn natural soil and catchment water storages are depleted. This low flow season, known from low elevated regions (Laaha and Blöschl, 2006), also leads to drought impact occurrences. The statistical tests provides proof that the summer season differed significantly from winter, but not from autumn.

Summer and often early autumn impact occurrences were dominated by the categories *Agriculture and livestock farming* and *Public water supply*. We indeed observed higher fractions of autumn impacts in the high-altitude region mostly related to *Public water supply*, followed by 'Reduced availability of irrigation water' (1.4). The relevance supports the expectation that water supply depends not only on direct precipitation, but also on natural water storages feeding springs used for drinking water. Reservoirs are likely managed differently across the Alpine Space, depending on the reservoir location and purpose. Hence, clarifying upstream-downstream dependencies would be a prerequisite to understand in more detail, why and where impacts have happened. Regarding the Alpine Space, autumn did not differ significantly from summer, thus highlighting the importance of this season for the European Alpine region.

Although we found least winter impacts, this season should not be neglected. Several studies show winter as an essential part of the droughts development, and suggest the delayed effects by summer and autumn accumulating in winter and winter as the early driver for upcoming impacts in the following seasons (van Loon et al., 2010; Livneh and Badger, 2020). Our compiled winter impacts differed slightly by their composition. The fraction of impacts related to *Agriculture and livestock farming* decreased (especially in the high-altitude and Southern region) with the last crop harvests in autumn. The impacts in the categories *Public water supply* and *Waterborne transportation* were also less in winter but with comparable fractions as in autumn. In contrast, impacts on *Tourism and recreation* peaked in winter, driven by the high-altitude region (all impacts from the southern region in this category were as well located in the high-altitude region). In the EDII$_{ALPS}$ most impacts in this category reported limited snow availability and snow production, both threatening ski tourism. Several studies raise the topic's attention, as drought harming tourism should not be ignored in management of snow tourism in mountain regions (Abegg et al., 2007; Gilaberte-Búrdalo et al., 2014; Spandre et al., 2019).

In almost all domains, more than 10 % of all impacts occurred in spring and the highest fractions was located in the Southern region (14 %). This can be related to the Mediterranean climate that is in general warmer and drier also in the early stage of the year causing an earlier start of the vegetation season. A corresponding example published in April, 2020 by an Italian newspaper is summarised as follows: "In the Brescian area, emergency irrigation was carried out on wheat, barley and fodder, but also on freshly sown corn. [...] In some cases the seedlings begin to dry out. Maize is also suffering: sowing took place between the end of March and the first days of April, but the lack of rainfall is compromising growth [...]. In some cases, farmers preferred to postpone sowing. Wherever possible, emergency irrigation is used [...]". The constitution of the EDII$_{ALPS}$' spring

impacts differed from that of the EDII$_{EU}$. We found substantially less impacts related to *Public water supply*. In mountain regions, the precipitation in winter does not evaporate as quickly as in lower elevated regions and soil moisture may replenish. In higher elevations, precipitation is first stored as snow and will not replenish the water storages before all snow is melted around July (both soils and artificial reservoirs) - or even later if glacier melt is used for filling water storages. Both processes likely mitigate water shortages in spring in mountain areas, leading to less D$_H$ impacts related to low discharge and groundwater storage. Furthermore, we found a higher fraction of *Forestry* impacts occurring in the Northern region. Delayed summer and autumn effects can persist over winter, especially in winters with temperatures cold enough to hinder precipitation to function as soil water, as it is stored in snow (van Loon et al., 2010). This can lead to dry soil and vegetation more vulnerable to drought. Additionally, most plants reduce their water intake during the cold season, to be less prone to frosts (Theocharis et al., 2012). Thus, cold winters do not prevent drought impacts in spring. The high fraction of impacts related to Tourism and recreation in the high-altitude region is mostly due to ski tourism lasting into spring.

The EDII$_{ALPS}$ reveales several delayed effects between impact categories. A delayed start and termination of D$_H$ impacts in all domains confirma the expectation that drought types occur in a particular order, which is not as clear in the EDII$_{EU}$. Further, all domains reported more D$_H$ impacts compared to D$_{SM}$ impacts underlying the assumption that drought impacts accumulated over time. Both effects were shown to be strongest in the high-altitude region. In the high-altitude region, snow accumulation in winter and in general lower air temperatures lead to better water availability in spring and early summer and subsequently, as shown by our results, less impacts (smallest fractions of impacts in spring and summer). Further, this typical mountain hydrology likely delays the D$_H$ impacts, as water released by snowmelt lead to longer water availability of natural storages typical for upstream areas. This effect could be found in all domains of the EDII$_{ALPS}$, but not in the EDII$_{EU}$. However, the high-altitude region experienced relatively the most D$_H$ impacts later in the year, showing the regions dependency on water and the need for management strategies also in upstream areas. The impact specific seasonalities in combination with spatial differences may need different seasonal indices in an impact-targeted drought monitoring and early-warning system across the Alpine region. Stagge et al. (2015) and Bachmair et al. (2016) presented different methods to model and predict impact occurrences with report-based datasets that could be applied to EDII$_{ALPS}$. In addition, to inform risk assessments at smaller scales a more complete spatial coverage of drought impact data collection is essential.

## 5 Conclusions

The presented EDII$_{ALPS}$ constitutes the first comparative view of drought impacts across the European Alpine region. The mountain specific database covering different countries may serve as an example how international collaboration can customise existing databases such as the EDII with a moderate effort to make them useful in regions that have previously not really been a focus. Mountain regions are usually not related to drought impacts due to their water-rich character. Nonetheless, our study presents the European Alpine region vulnerable to the hazard of drought. The EDII$_{ALPS}$ archives a great amount of impacts mostly related to agriculture and livestock farming followed by public water supply. These affected sectors are firmly established in the region, wherefore adaption and management strategies are essential for the future climate regimes. Apart

from the most relevant sectors, we present a surprising diversity of impacts covering a wide range of environmental, economic and societal effects that confirm the multifaceted character of drought in the EDII$_{ALPS}$. This growing diversity over time is likely due to the increasing complexity of the socio-economy across the European Alpine region with various sectors exposed and/or vulnerable to drought. In addition, the number of impacts increases substantially over time.

Key characteristics of drought impacts in the region are that impacts mostly occur in summer and early autumn regardless of region, climatic condition or altitude. Typical to EDII$_{ALPS}$ are also some winter impacts related to ski tourism while spring impacts occur mostly in the southern region. The regions' specific snow accumulation in winter likely mitigates water shortages through snowmelt contributions in spring and early summer. Further, this study proves the possibility to link impacts to hydrological drought and soil-moisture drought in order to analyse drought specifics in different hydrological and climatic conditions. For the mountainous regions, we could demonstrate the delay between impacts classified as related to soil-moisture drought and those classified as related to hydrological drought. All these seasonal effects of water redistribution and demand are essential to understand drought as a hazard across different climatic and altitudinal zones. Therefore, our study presents a good starting point from the impact perspective. Despite some biases in the current database, the amount of impacts we compiled in the EDII$_{ALPS}$ should raise awareness. Future climate predictions with increased drought severity, less snow and shift in precipitation patterns, suggest the European Alpine region will be further impacted by drought. For drought risk assessments and management strategies EDII$_{ALPS}$ will have to be analysed and adapted further by more complete spatial coverage on the local scale and by modelling impact occurrence to predict the future potential of drought impacts.

*Data availability.* The EDII$_{ALPS}$ V1.0 (last update 8th of January, 2021) is available under 10.6094/UNIFR/218623.

*Author contributions.* Ruth Stephan, Mathilde Erfurt and Kerstin Stahl, designed the research. All co-authors provided data. Ruth Stephan moderated all EDII$_{ALPS}$ entries, carried out the analysis and created the graphs, maps and tables in the manuscript. Ruth Stephan prepared the manuscript with contributions from Mathilde Erfurt and reviews from all co-authors.

*Competing interests.* The authors declare that they have no conflict of interest.

*Acknowledgements.* This research was funded by the EU Interreg Alpine Space Programmes project ADO (Alpine Space Observatory) with the project number ASP940. The article processing charge was funded by the German Research Foundation (DFG) and the University of Freiburg in the funding program Open Access Publishing. We acknowledge the access to the European Drought Impact Report Inventory database, which has initially been funded by the EU FP7 project DROUGHT-R&SPI. We thank especially Veit Blauhut who enabled the access. We also acknowledge the Provincia autonoma di Bolzano-Alto Adige – Ripartizione Innovazione, Ricerca e Università as financing

institution of the AquaMount project. We acknowledge Stanka Klemenčič and Mojca Hribernik from the Slovene Chamber of Agriculture and Forestry for their effort to collect Slovenian drought impact data.

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
