# Peer review of "An inventory of Alpine drought impact reports to explore past droughts in a mountain region"

_Natural Hazards and Earth System Sciences, 2021_

## Referee Comment (RC1)

**Review of Stephan et al. An Alpine Drought Impact Inventory to explore past droughts in a mountain region**

*By Anne Van Loon*

This is a very interesting paper, presenting a drought impact database for the Alpine region and comparing the reported impacts with a similar European drought impact database. The authors also show interesting patterns of drought impacts in space and time. I really like the figures, they condense a lot of information in a few very clear maps and graphs. I have a few minor suggestions for improvement of the paper. I summarise these first and then share my more detailed comments below.

One comment is on the region you are looking at. You need to be clear from the start about the area you are studying. Now you mention 'Alpine Space', European Alpine region, mountain-foreland region, mountain-to-foothill transitions, etc. 'Alpine Space' is now defined in the Methods section, but it should be explained earlier in the Introduction.

My second point is on the compilation of the database. You need to explain more on how the information from the different sources were collected. Have all these data sources been compiled before and are they publicly available? Or have you collected, translated and mined text-based reports yourself?

Thirdly, please explain how you have dealt with overlap between EDIIeu and EDIIalps. Did you include EDIIeu impact reports for your region into EDIIalps? Or did you include your EDIIalps in the EDII database? It would be very interesting to compare your EDIIapls with the original EDII entries for your Alpine region to see the effect of differences in data collection.

Finally, I found it surprising that there is not many impacts reported in the Energy category. Please expand the discussion on this. Also in the Discussion, I expected you to say more about how perceptions of drought might have influenced the results, for example for the Energy category and for the impacts in the southern part of the Alpine region.

**Specific comments:**

**Abstract:**

- What is 'Alpine Space'?
- "The amount of more than 3200 compiled reports on negative drought impacts demonstrates the need to move from emergency actions to better preparedness" > not sure if the amount of impact reports demonstrates this need, maybe rephrase?

**Introduction:**

- Socio-economic drought: It is a bit confusing how this is different from the impacts that you are investigating. For example the sentence: "These indirect impacts are the least tangible and often related to DSE." (And in the Methods: "DSE is challenging to relate to specific impacts") Please remove DSE as a drought type to avoid confusion between hazard and impacts?
- P.3 l.58: maybe also mention DH impacts on hydropower production. A quite important sector in the Alpine region I thought. But from your results I see that the Energy and industry category is not often reported. Do you know why? Is it not an issue or does this just not end up in the newspapers?

**Methods:**

- Fig. 1: can you indicate the countries and their borders in the map? This is important as later in Fig. 2 & 3 you report impacts by country.
- Grouped into four domains: Northern, Southern, high-altitude and pre-Alpine. Please make it clear in the text that these are two different subdivisions. So you actually grouped two times into two domains. This comes back in the Results section (see below).
- P.4 l. 98: explain how you defined Europe as comparison for the Alpine Space analysis.
- P. 5 l.106-110: Are all the sources listed publically available? Or did you also compile some of these yourself, for example the Italian and German text-based reports? What is the origin of these text-reports?
- How did you check for overlap with the existing EDII? Did you include your EDIIalps in the EDII database, or the other way around did you include original EDII entries in your EDIIalps database? Did you compare your EDIIalps with the EDII entries for your Alpine region to see the effect of differences in data collection?

**Results:**

- Fig. 2: I would suggest to add the distribution of reported impacts in the Alpine Space but based on the EDIIeu, so that you can compare with those in your EDIIalps. Also, maybe refer to the figure a bit more in the text to help the reader.
- P. 7 l. 185-186: here you need to be careful with the regions again. You can only compare between the regions within a subdivision, so Northern with Southern and pre-Alpine with high-altitude, so rephrase to "… more reports located in the Northern than in the Southern region and more in the pre-Alpine than in the high-altitude region".
- Table 1: Add in the caption that this is a subset / selection of tested combinations. Now this is a bit hidden in the table footnote. In general extend the caption so that it is clear which results are presented in the table.
- P.7 l. 191-193: Is there a mistake in the numbers or am I failing to understand the difference between the numbers? For EDIIeu the numbers of the two categories add up to 56%, but for EDIIalps 48 + 21 = 69% and not 61%.
- P.10 l.38-39: "the extreme relative dominance of the category Agriculture and livestock farming in 1976 (82 %) decreased substantially in the years 2003, …" rephrase to something like "the category Agriculture and livestock farming was extremely dominant in 1976 (82 %) and much less apparent in the years 2003, …", because there is not necessarily a change over time.
- P.11 l.256: Alpine Space
- Table 2: Like in Table 1, extend the caption so that it is clear which results are presented in the table.
- Figure 5: In the figure, Dsm and Dh are represented as SMD and HD. Please change for consistency and clarity.

**Discussion:**

- P.15 l.346: "30 % of all impact data across Europe is located in our study region" > Do you mean that within EDIIeu 30% of the impact report is located in your Alpine Space? Or that your EDIIalps has 30% of the impact reports of EDIIeu? The latter is not a conclusion you can draw because of the differences in data collection.
- P.16 l.378: "ressort"? Do you mean report?
- Please discuss the relatively low amount of impacts in the category Energy (and Tourism).

- P.17 l.413-414: Or maybe because dryness is more normalised in a Mediterranean climate and water shortages are not always reported as drought impacts?
- P.19 l.453-455: Or water shortages because of a delayed or lower snow melt might not be reported as drought impacts?

**Conclusions:**

- The first paragraph (l.475-487) fits better in the Discussion section than in the Conclusions.

---

## Referee Comment (RC5)

**Review: An Alpine Drought Impact Inventory to explore past droughts in a mountain region. Stephan et al. NHESS-2021-24**

This was a really nice paper describing the development of a new drought impact inventory for the Alpine region. The graphs and analysis used were novel compared to other papers which describe and analyse the EDII. I suggest only minor revisions to the paper before publication, these are discussed below and some more specific comments on the figures, text and written English are listed below.

Did you consider splitting (or aggregating the NUTS regions, perhaps NUTS3) by basin – this could be interesting in Section 3.1 where you look at the spatial differences in impacts, as well as Section 3.4 where you consider the different drought types.

Section 2.2 – further information on the specific sources of impact data for the Alps EDII would be useful. It's not clear for example, whether the Italian and German text reports were from newspapers, government reports etc. and it would be useful to have more information on the Propluvia French data as it is not clear what it is. Please include URLs to the sources where appropriate.

Section 2.2 - I suggest that you could put the example impact reports in a table – this would be easier to read and for readers to see the differences between the impact data from the different sources.

L149 – the choice of the case study years has not been explained – it would be good to introduce these years in the introduction perhaps with appropriate references in the introduction e.g. Lahaa et al. 2017 (https://doi.org/10.5194/hess-21-3001-2017)

L193 – The percentages here do not match up with those in the previous sentence – please check these figures

L387-389 – Here you say that 2003, 2015 and 2018 depicted are more comparable picture, but in the following state there was a remarkable difference in 2003. These points seem to contradict each other – unless the comparable picture refers only to the Agriculture and Forestry categories. Please clarify.

Section 4.3 – it would be interesting to consider the temporal trends and drought occurrence in the context of other known drought events (aside from the case study events used in the paper). For example, comparing the results to papers that consider the timing, propagation and characteristics of drought events (e.g. Laaha and Van Loon 2015 (https://doi.org/10.1016/j.jhydrol.2014.10.059), Haslinger & Bloschl 2017 (https://doi.org/10.1002/2017WR020797) such as Spinoni et al 2015 (https://doi.org/10.1016/j.ejrh.2015.01.001), Sheffield et al. 2009 (https://doi.org/10.1175/2008jcli2722.1) and others)

You make an interesting comment in the final line of the paper on the application of the EDII ALPS data; it would be interesting to explore this further in the discussion.

The written in English in places could be improved as in some cases it was difficult to really understand what was meant. Some points on this are listed below.

**Figures and tables**
**Figure 1** – adding the country boundaries would be helpful here especially as you use the countries in the analysis later on. In the right hand figure there seems to be a small region inside the high

altitude region that isn't labelled. It is also quite hard to see the NUTS boundaries (which are also not labelled – you could consider adding them to the key) it might help to make the maps bigger and the boundaries thicker so they stand out against the elevation.

**Table 1** – foot note 1 should be moved to the caption for clarity (and could be mentioned in the text). You could consider showing these results in a heat map of all the pairs, highlighting these significant pairs (same for Table 2).

**Figure 2a** – please add a legend for the grey shading of the NUTS regions and make the country borders clearer

**Figure 2b** – in the caption please explain that data are shown for each sub-category so it is clear why for example the southern region has two labels for the agriculture impacts and why there are faint grey lines within each impact category block

**Figure 3** – it could be the resolution of the figure in this draft version, but the red text is quite hard to read – it is also not explained what the p value is in the caption.

**Figure 3** – I guess that the dotty plot is the 'counts of all reports per country and year' - please add a legend to indicate what size of the circles mean. I also suggest you label this figure 3b and the current figure 3b, to 3c

**Table 2** – I assume the dashes in the rows for the high altitude and southern regions indicate that there were no significant pairs for these regions, please clarify this in the caption.

**Figure 4** – the dates for the seasonal summaries start in March for spring, you could consider doing the same for the time series plots so that the values for the winter are all together.

**Figure 5** – the caption uses the acronyms DSM and DM but in the plot these are labelled as SMD and HD, please make these consistent here (and throughout the paper – sometimes you use the acronym and sometimes not).

**Figures 4 & 5** – In the captions for these two figures the sentence "Monthly values are related to frequency of the month with most impacts." isn't very clear - does this mean that for each impact category the monthly data points for each month are from the year with the most impacts in that category? Please clarify this.

**Minor points**

L72 – you could introduce the acronym $EDII_{ALPS}$ here

You introduce the abbreviations DSM and DH ~L40-45 but don't always use them, for example in the figures and in the discussion section.

L90 – should the Eurostat reference be a full reference with a date?

L142 & Data availability – doi to be updated if possible

L146 – it's not clear what total numbers you are referring to

L165 – it would be useful to refer to Table S1 here - you don't directly refer to it until much later

L169/170 – and throughout, you are not so consistent with the use of your acronyms, so here for example you could use EDII ALPS and EDII EU instead of the Alpine Space and the entire European region (you also don't define what the European region is – could you show on the inset map in Fig 1?)

L178 it's not clear what you are referring to by 'groups' here and L180 the end of this sentence is not clear – what do you mean 'if we tested more than two'?

L235 – it is not clear which three years you are referring to here

L256 – do you mean that the summer was always significantly different to winter?

L257 – you mention that summer and autumn were not significantly different in terms of the impacts, how does autumn compare to the other seasons?

L377-381 – it could be useful to link to Stahl et al. 2016 (https://doi.org/10.5194/nhess-16-801-2016) here which discusses the biases of the EDII and the text based impact report approach

L433 – you say you presented the least winter impacts, do you mean to say that there were fewer impacts in winter? The word presented here implies that there were impacts but you didn't show them

L442 – "as the problem is real and should not be ignored in management" this is quite informal and the wording could be improved

 L454 – is there a word missing at the end of this sentence? What is the glacier melt used to fill? (presumably lakes?)

**English/grammar/spelling**
Some points on spelling/the written English are given here:

Check tenses throughout – you mix between present and past tense, particularly in the discussion

L5 – is this missing 'report' from "Alpine Drought Impact Inventory" i.e. to be consistent with the EDII (European Drought Impact report Inventory)?

L8 – suggestion to change 'to entire Europe' (and similar phrasings throughout, e.g. L10) to 'the whole of Europe' to improve the grammar

L36 – 's' missing from end of Alps

L60 – add a comma after "In mountainous regions" to improve the readability of the sentence

L64 – should 'report' in European Drought Impact Report Inventory be lower case?

L65 – when talking about the 'Tourism and recreation' impact category throughout the paper, Tourism is spelt wrong ('Tourims')

L70 – needs either 'data' (or similar) at the end of the sentence or change to 'the EDII' so that it makes sense

L97 - suggested change: we chose the spatially higher resolved NUTS 3 regions because → we chose NUTS3 regions with a higher spatial resolution because

L102 – missing 'The' at the start of the sentence which currently starts "EDII itself"

L117 – I think here exemplary should read exemplar

L150 – should the colon should be a full stop?

L218 – where you have used the word 'relevance', do you mean 'occurrence' (applies to some later occasions where the word relevance is used e.g. L427)

L421  - the word 'especially' is not clear – do you mean that there is an expectation that drought impacts tend to occur in the summer?

L425 – suggested change: Summer and often early autumn impact dominance most clearly shown for the impact categories → Summer and often early autumn impact occurrences were dominated by the Agriculture and livestock farming and Public water supply

Some sentences were hard to follow and could be improved for example: L97-99, L453-454

---

## Author Comment (AC1)

**Reply to Anne Van Loon**

We would like to thank Anne Van Loon for her constructive comments and feedback on this manuscript. We think that the suggested revisions based on the Referee's comments will certainly improve the article. Please find our responses (in blue) to the main points raised (shown in black) below.

One comment is on the region you are looking at. You need to be clear from the start about the area you are studying. Now you mention 'Alpine Space', European Alpine region, mountain-foreland region, mountain-to-foothill transitions, etc. 'Alpine Space' is now defined in the Methods section, but it should be explained earlier in the Introduction.

➜ We agree that this needs to be more clear. See detailed answers below.

My second point is on the compilation of the database. You need to explain more on how the information from the different sources were collected. Have all these data sources been compiled before and are they publicly available? Or have you collected, translated and mined text-based reports yourself?

➜ A mix of both. We comment more on the availability of the original sources below. And we will make the specific classified collection that we used here available through a repository. We can only ask for the doi once we know that the paper will be accepted and will then include this in the final revision. We agree the source data and its compilation needs clarification. See details below.

Thirdly, please explain how you have dealt with overlap between EDIIeu and EDIIalps. Did you include EDIIeu impact reports for your region into EDIIalps? Or did you include your EDIIalps in the EDII database? It would be very interesting to compare your EDIIapls with the original EDII entries for your Alpine region to see the effect of differences in data collection.

➜ This comment follows the previous one and will be clarified as part of it.

Finally, I found it surprising that there is not many impacts reported in the Energy category. Please expand the discussion on this. Also in the Discussion, I expected you to say more about how perceptions of drought might have influenced the results, for example for the Energy category and for the impacts in the southern part of the Alpine region.

➜ We were surprised as well. There are some possible explanations that we will improve in the discussion. See details below.

**Specific comments:**

**Abstract:**

What is 'Alpine Space'?

➔ The EU Interreg-Programm Alpine Space introduced the spatial extent 'Alpine Space' which "covers the Alps and their foothills, as well as different climatic zones", as we defined in section 2.1. We see that this definition comes late in the text and will therefore refer to the study region as "Alpine region" in the abstract.

"The amount of more than 3200 compiled reports on negative drought impacts demonstrates the need to move from emergency actions to better preparedness" > not sure if the amount of impact reports demonstrates this need, maybe rephrase?

➔ We agree that the conclusion should incorporate more arguments than solely the amount of negative drought impacts. We will rephrase the sentence so that our study demonstrates drought as a hazard that requires attention in the Alpine region.

**Introduction:**

Socio-economic drought: It is a bit confusing how this is different from the impacts that you are investigating. For example the sentence: "These indirect impacts are the least tangible and often related to DSE." (And in the Methods: "DSE is challenging to relate to specific impacts") Please remove DSE as a drought type to avoid confusion between hazard and impacts?

➔ We see the confusion and will remove DSE.

P.3 l.58: maybe also mention DH impacts on hydropower production. A quite important sector in the Alpine region I thought. But from your results I see that the Energy and industry category is not often reported. Do you know why? Is it not an issue or does this just not end up in the newspapers?

➔ We also expected this category to be more often reported, as it is an important sector in the Alpine region. We will check some more sources but reasons (anecdotally suggested by stakeholders) include a stronger dependence on the energy market than on water. Also, most hydropower is produced from reservoir storage and most profitably sold for peak demand, whereas run-off-the-river hydropower sold for base demand financially plays less of a role for the (diversified) producers. We can elaborate a bit more on the issue.

**Methods:**

Fig. 1: can you indicate the countries and their borders in the map? This is important as later in Fig. 2 & 3 you report impacts by country.

➔ Yes, we will modify the map in order to show as well the countries, respectively part of the countries in the Alpine Space. We propose to show the borders in the additional overview map. Otherwise the overlapping lines of the countries, NUTS 2 and 3 regions, and subregions are not visible separately. See our proposed example (Figure R1):

[Figure]

Figure R1: The "Alpine Space" study area within Europe (a) for which the Alpine Drought Impact Inventory (EDII_ALPS) was developed, showing the paired subregions for the analysis: (b) the Northern and Southern region divided by grouped NUTS 2 regions, and (c) the pre-Alpine and high-altitude region divided by grouped NUTS 3 regions.

Grouped into four domains: Northern, Southern, high-altitude and pre-Alpine. Please make it clear in the text that these are two different subdivisions. So you actually grouped two times into two domains. This comes back in the Results section (see below).

➔ Correct, we see that our phrasing is misleading and we need to clarify this grouping into different regions better in the revised version.

P.4 l. 98: explain how you defined Europe as comparison for the Alpine Space analysis.

➔ Please read our response to the following point for our answer.

How did you check for overlap with the existing EDII? Did you include your EDIIalps in the EDII database, or the other way around did you include original EDII entries in your EDIIalps database? Did you compare your EDIIalps with the EDII entries for your Alpine region to see the effect of differences in data collection?

➔ We agree that our explanation of how we defined the different regions was not clear enough and thus raised your questions. We will rephrase the parts describing how we updated EDII and subsetted respectively defined the different regions.
To clarify for further comments:
1. We considerably updated the original EDII database (a) with sources that had not been investigated before (Unwetterchronik in AT, Propluvia in FR), (b) with several other reports we compiled ourselves (especially German and Italian text-based reports), and with sources that had been used previously by EDII, but which did not receive an update for the more recent years (Drought.ch, DCMSEE). The updated version is called EDIIeu throughout the preprint.
2. We then subsetted the reports located in the Alpine Space from EDIIeu and called this EDIIalps, which is thus a part of EDIIeu.
3. We further split EDIIalps two times to compare different climatic and altitudinal conditions: Northern vs. Southern region and pre-Alpine vs. high-altitude region.

P. 5 l.106-110: Are all the sources listed publically available? Or did you also compile some of these yourself, for example the Italian and German text-based reports? What is the origin of these text-reports?

➜ The data we compiled were all publicly available. They stem from different sources meaning that some of them have been compiled before for other overviews or databases, such as the Unwetterchronik and the DMCSEE bulletins. We classified and transferred them into the EDII system. We collected the Italian and German text-based reports ourselves by searching News print and other media etc.and also classified and entered these into the EDII system. We specified the information sources to keep track of the origin of the text-reports. That means that we have the possibility to trace back, if the original source was a research article, newspaper article, governmental report, an entry in another database and so forth. This way we kept the standards the European Drought Impact Report Inventory (EDII) requires.

**Results:**

Fig. 2: I would suggest to add the distribution of reported impacts in the Alpine Space but based on the EDIIeu, so that you can compare with those in your EDIIalps. Also, maybe refer to the figure a bit more in the text to help the reader.

➜ The distribution of reported impacts in the Alpine Space is based on EDIIeu meaning that EDIIalps is a subset of EDIIeu (see our response in the Methods section). We will integrate the Figure more in the text.

Thank you for the other minor comments for the result section. We agree with the comments and suggestions and will address all minor points in the revised manuscript.

**Discussion:**

P.15 l.346: "30 % of all impact data across Europe is located in our study region" > Do you mean that within EDIIeu 30% of the impact report is located in your Alpine Space? Or that your EDIIalps has 30% of the impact reports of EDIIeu? The latter is not a conclusion you can draw because of the differences in data collection.

➜ We mean that 30 % of all impact data in EDIIeu is located in the Alpine Space, which we call EDIIalps (see our explanation to your question in the method part). We see the need to clarify the overlap between EDIIeu and EDIIalps and will add an explanation.

P.16 l.378: "ressort"? Do you mean report?

➜ We did not mean "report", but the federal forest institutions in Germany that regularly publish assessments about the forest conditions. We will rephrase the sentence to mitigate misunderstandings with the word "ressort".

Please discuss the relatively low amount of impacts in the category Energy (and Tourism).

➜ We also expected the categories Energy and tourism to be more reported. For the energy sector, please see our comments above about the dominating role of energy price and market. In general, drought is rarely associated with cold winter droughts, but Alpine tourism is mainly affected from winter droughts. This could be the reason for the relatively low amount of impacts. In fact, our project group also expected tourism to be more present in reports. There may be reasons such as an overlap low economic relevance nationally or in those places where reports are made, the language used to

refer e.g. to economic losses due to low snowpack for winter sports and a lack of a verbal link to "drought" (and related terms), which the EDII guidelines require to be made by a report used to create an impact entry. We will discuss this point in more detail in the revised manuscript.

P.17 l.413-414: Or maybe because dryness is more normalised in a Mediterranean climate and water shortages are not always reported as drought impacts?

➔ This might be an explanation for regions in the Southern parts of the Alpine Space. For other areas in the Alpine Space it could be that "cold winter droughts" are not associated with water shortages and therefore are not reported as drought impacts. We will further discuss this point in the revised manuscript.

P.19 l.453-455: Or water shortages because of a delayed or lower snow melt might not be reported as drought impacts?

➔ We think your raised point is especially important in the mountainous regions. Not only are some drought impacts delayed in the Alpine region, they also can occur in another region (upstream, downstream) and therefore might not be associated with drought. We will elaborate this point in the discussion of the revised manuscript.

**Conclusions:**

The first paragraph (l.475-487) fits better in the Discussion section than in the Conclusions.

➔ We also had this discussion beforehand and agree to rearrange this.

---

## Author Comment (AC4)

**Reply to Referee 2**

We would like to thank you for your constructive comments and feedback on this manuscript. We think that the suggested revisions based on the Referee's comments will certainly improve the article. Please find our responses (in blue) to the main points raised (shown in black) below. We ordered the specific comments to the main sections of the paper in order to supply the same structure to all reviews.

Both the data retrieval and the characteristics of the dataset need to be described further. For instance, the period covered by the dataset should be clearly stated in the text and the relationship between the EU and the Alps dataset in terms of counts and repeated entries should be explicitly addressed. I agree with Reviewer 1 that it would be interesting to describe and discuss the difference found for the overlap region (i.e. Alpine NUTS in the EDIIeu dataset and the EDII Alps dataset) as a result of the new impact retrieval process.

→ Corresponding to our answer to Anne van Loon, we agree that this needs to be more clear. We will rephrase the parts describing how we updated EDII and subsetted respectively defined the different regions. Further we will clearly add the period covered by our dataset. Regarding the effect of our impact retrieval process we reran the analysis with a previous version of the database (i.e. status from September 2019) and suggest that we will elaborate further the effect of our update on the presented analysis and then decide how to include it in the manuscript or provide it as additional information in the Supplementary Material.

It would be useful to strengthen the discussion in terms of the policy relevance of the database and of the trends that emerged from the data analysis. How could policy makers use them? What type of decisions could they inform?

→ Foremost we consider our contribution research work that will have to be analysed and adapted further to be used by practitioners and/or policy makers. However we can add a more elaborated paragraph in the discussion or conclusion about ways towards applications based on the relevance of our data and findings. These may include that the two seasonally differently occurring impact groups and the different regions may need respectively different seasonal indices in an impact-targeted drought monitoring and early-warning system across the Alpine region. More generally, we may highlight the benefit to implement systematic drought impact information/data collection to inform necessary risk assessments at smaller scales or for more complete spatial coverage.

The manuscript presents and analyzes a large body of data and it is always challenging to present large datasets and complex patterns. In some paragraphs I had some difficulties following the text. I have noted down the sentences that I found particularly challenging.

→ Thanks for pointing that out. We will improve the sentences you noted so that the reader can follow our ideas.

**Specific comments:**

→ Thanks for the notes to all sentences or paragraphs you suggest to rephrase. We will go through them and improve the phrasing.

**Introduction:**

p. 2 line 49 Please check the definition of socioeconomic drought. I would expect it to be "insufficient water availability to meet the ordinary demands of society and economic activities" (now it says "inadequate supply of some economic good…").

→ According to the suggestion by Anne van Loon, we would like to remove the drought type DSE. Then, we would also not need to discuss the definition of this drought type further.

p.3 Please check the leading questions and may sure that they can be clearly differentiated. The first and the third one seems very similar to me.

→ The first question deals with drought impacts in the Alpine mountains and in the Alpine regions compared to drought impacts in whole Europe. In contrast, the third question deals with the distribution of the different impact *types* in the Alpine space. We especially focus on the question if a specific impact type (e.g. impacts caused by hydrological drought), occurs in a specific season. We will clarify the difference in the revised version.

**Methods:**

p.3-4 Please specify the altitude ranges used to delineate the different spatial domains and based on what they were defined.

→ On p. 4 line 95 we defined the high-altitude region and the pre-Alpine region as follows:

*(3) The "high-altitude region" identified with NUTS 3 regions for which ≥ 30 % of the area are higher than 1000 masl versus (4) the "pre-Alpine region" covering all remaining NUTS 3 regions.*

p. 5 Please specify in the text the time period covered by the drought impact search. IT would be useful also to know more about the search process: did the authors use a search by key words? If so, what words did they use and how effective the search was?

→ EDII first impact report goes back to 1448. This very historical information for southwestern Germany was retrieved from the collaborative research environment tambora.org (https://www.tambora.org/; Glaser et al., 2015; Glaser, 2013). However, most collected reports stem from the late 20th century as shown in Fig. 3. We applied the same search method as described in Stahl et al. (2016) in order to be consistent and will add this information in our revised manuscript.

p. 6 line 167 "loess": do you mean "loss"?

→ According to the applied method, we mean "loess".

p. 7 line 183: Do the events identified in this new search partially overlap with the EDIIeu ones? How many about of the 3,200 are also counted among the 10,600 ones?

→ As EDIIalps is part of EDIIeu all of the reports within the Alpine Space region are also counted for EDIIeu. For further details see our response to Anne van Loon:

*We agree that our explanation of how we defined the different regions was not clear enough and thus raised your questions. We will rephrase the parts describing how we updated EDII and subsetted respectively defined the different regions.*
*To clarify for further comments:*
1. *We considerably updated the original EDII database (a) with sources that had not been investigated before (Unwetterchronik in AT, Propluvia in FR), (b) with several other reports we compiled ourselves (especially German and Italian text-based reports), and with sources that had been used previously by EDII, but which did not receive an update for the more recent years (Drought.ch, DCMSEE). The updated version is called EDIIeu throughout the preprint.*
2. *We then subsetted the reports located in the Alpine Space from EDIIeu and called this EDIIalps, which is thus a part of EDIIeu.*
3. *We further split EDIIalps two times to compare different climatic and altitudinal conditions: Northern vs. Southern region and pre-Alpine vs. high-altitude region.*

**Results:**

p. 8 line 225-226: please rephrase the sentence "Thus, the frequency …" (difficult to follow)

→ We agree and will rephrase it.

p. 9: Could you please explain the usefulness of comparing the NUTS 2 regions (Table 1)? What information does this comparison provide?

→ We wanted to compare not only the national parts of the Alpine Space, but as well smaller regions, as the mountainous terrain is very heterogeneous. Therefore, we included the comparison of between the NUTS 2 regions. In Table 1 we show the several NUTS 2 regions located in Italy differed to NUTS 2 regions in Austria, Switzerland and Germany. An effect we did not see in the comparison between the countries.

p.10 lines 238-252 the text is difficult to follow. I recommend simplifying it.

→ We agree and will rephrase it.

**Discussion:**

p.14 the authors conclude that the chosen data sources proved to be suitable as impacts were clearly liked to the drought occurrence. I suggest revising this statement: the data collection was set up to detect only impacts that are explicitly linked to drought in reports that are being searched, so it is no surprise that the retrieved impacts met that requirements. Instead, the authors could discuss (or at least mention as a caveat/limitation) to what extent they may have missed drought impacts that were not explicitly linked to drought in the reports.

→ We agree with you that the statement needs revision and we will do that.

p.16 line 364, I think "common" (or similar word) is missing between "most impact".

→ Thanks. We will revise it to "The second most frequent impact category…"

p.16 line 370-372: please rephrase the sentence starting with "Whether upstream", it is difficult to follow.

→ We agree and will rephrase it.

p.17 line 414: why would impacts in the Southern region be "too local"?

We agree that this phrasing is not precise enough and requires clarification. What we meant is that due to the better coverage of the Northern Region with impact reports and hence 'data', we assume a more regionally complete representativeness. In the Southern Region, impact report data is more scarce and hence may have gaps in the spatial representation.

p.18 line 445: the text says that the Southern region reported the most impacts in spring while on the same page, on line 417, it is said "summer and early autumn are the seasons with the most drought impacts in all domains". Please clarify this apparent contradiction.

→ Thanks for pointing that out. We see that this phrasing raises questions. On line 417 we state that "summer and early autumn are the seasons with the most drought impacts in all domains" supported by Figure 4. The total counts of the reports for each season show that the most impacts were reported in summer and autumn for all plots (a) - (f). If we compare the Southern region with the other regions within the Alpine Space, then this is the region reporting relatively the most (14 % of all reports stem from spring). We will rephrase the sentence on line 445 to clarify this.

p.19 line 460 please rephrase (unclear sentence)

→ We agree and will rephrase it.

p.19 lines 465-466 Please rephrase (unclear sentence)

→ We agree and will rephrase it.

**Conclusions:**

p.20, line 483: the conclusion "impact data collection EDII alps is therefore shaped by national priorities and societal effects "is unclear. Also the recommendation about customization EDII (lines 485.487) require some more elaboration in terms of what that "customization" would be.

→ We agree that the sentence is unclear and will rephrase it. What we wanted to express is that the data collection might be influenced by the different national foci and different collection efforts.

p.20 Line 488: I recommend to rephrase the sentence starting with "our study…" as the fact of being water rich does not make a place not vulnerable to drought.

→ We agree with you that the natural hazard of drought can occur everywhere, since drought is defined as a deviation from normal. However, drought impacts as an expression of vulnerability or exposure or just as an issue of public awareness are not typically associated with the entire Alpine region. We will consider a more nuanced rephrasing and might add a reference.

p.20 line 493-494: please elaborate on the idea of the growing diversity of impacts over time. Is it really due to an increasing complexity of the socioeconomic system in the Alpine Space? Beyond the use of water to produce snow in ski resort, I would expect all the other uses and sectors affected by impacts in 2018 to exist and be well established also in the 1970s and later.

→ We agree that the increase in diversity of impacts may warrant a bit of weighing of possible causes for this observation that in fact we cannot disentangle easily. One aspect

which influences the database is the increase in information and access to information in general and this will be reflected in the breadth of impacts. A real growth in diversity of impacts over time, however, can also not be excluded as an explanation. Winter tourism is not the only sector which has changed since 1976. There was also an increase in population, summer tourism, water use etc. In the Alpine forelands the agriculture sector changed a lot and in the Alpine regions new infrastructure for energy and water use was established and subject to more market competition.

p.20 lines 498-501. As it is written now, the reader could think that the authors have compared the impact patterns with actual precipitation patterns or drought indices. It is my understanding that this comparison is beyond the scope of the paper. Instead, in section 4.3 the authors made an interesting attempt to explain the occurrence of impacts throughout a generic year based on the literature. I recommend rephrasing these lines to make sure that they reflect the actual content of paper's analysis.

→ We agree and suggest rephrasing: "For the mountainous regions, we could demonstrate the delay between impacts classified as related to soil-moisture drought and those classified as related to hydrological drought."

p.20, line 501-02. Please rephrase the sentence "all these…starting point" (unclear)

→ We agree and will rephrase it.

References

Glaser, R.: Klimageschichte Mitteleuropas: 1200 Jahre Wetter, Klima, Katastrophen, 3. Auflage, WBG Wissenschaftliche Buchgesellschaft, Darmstadt, 264 pp., 2013.

Glaser, R., Riemann, D., Kellersohn, A., Lentz, S., Hanewinkel, C., Beck, A., Vogt, S., Borel, F., Sidawi, W., Kahle, M., Vogt, J., Steller, H., Specht, S., and Koslitz, S.: Tambora – the climate and environmental history collaborative research environment, FreiDok plus, Albert-Ludwigs-Universität Freiburg, Freiburg, https://doi.org/10.6094/tambora.org, 2015.

Stahl, K., Kohn, I., Blauhut, V., Urquijo, J., De Stefano, L., Acácio, V., Dias, S., Stagge, J. H., Tallaksen, L. M., Kampragou, E., van Loon, A. F., Barker, L. J., Melsen, L. A., Bifulco, C., Musolino, D., de Carli, A., Massarutto, A., Assimacopoulos, D., and van Lanen, H. A. J.: Impacts of European drought events: insights from an international database of text-based reports, Nat. Hazards Earth Syst. Sci., 16, 801–819, https://doi.org/10.5194/nhess-16-801-2016, 2016.

---

## Author Comment (AC5)

**Reply to Referee 3**

We would like to thank you for your constructive comments and feedback on this manuscript. We think that the suggested revisions based on the Referee's comments will certainly improve the article. Please find our responses (in blue) to the main points raised (shown in black) below. We ordered the comments to the main sections of the paper in order to supply the same structure to all reviews.

**Introduction:**

Figure 1 – adding the country boundaries would be helpful here especially as you use the countries in the analysis later on. In the right hand figure there seems to be a small region inside the high altitude region that isn't labelled. It is also quite hard to see the NUTS boundaries (which are also not labelled – you could consider adding them to the key) it might help to make the maps bigger and the boundaries thicker so they stand out against the elevation.

 $\rightarrow$  We will improve Figure 1 and follow your and Anne van Loons suggestions. We will add a better schematic overview to clarify the different domains covered by EDIIeu and EDIIalps. See our suggested Figure R1.2 below. We would like to avoid adding the label of all NUTS regions, as the maps get very noisy and less clean.

**Figure R1.2.** (a) The  $\text{EDII}_{\text{EU}}$  covering the "Alpine Space" study area for which we updated the original EDII. The Alpine Drought Impact Inventory ( $\text{EDII}_{\text{ALPS}}$ ) covering the different Alpine countries and integrated in the  $\text{EDII}_{\text{EU}}$ . Our study region in its paired subregions for the analysis: (b) the Northern and Southern region divided by grouped NUTS 2 regions, and (c) the pre-Alpine and high-altitude region divided by grouped NUTS 3 regions.

**Methods:**

Did you consider splitting (or aggregating the NUTS regions, perhaps NUTS3) by basin – this could be interesting in Section 3.1 where you look at the spatial differences in impacts, as well as Section 3.4 where you consider the different drought types.

 $\rightarrow$  Yes, we also played with this idea and currently have a Master thesis working on this topic. The preliminary results of this thesis present differences between the basins of Rhine, Danube, Po, and Rhone, which show similar patterns as our results for the Alpine countries in Fig. 2. Due to the additional methodological steps, but more or less similar results, we decided not to include this in the manuscript.

Section 2.2 – further information on the specific sources of impact data for the Alps EDII would be useful. It's not clear for example, whether the Italian and German text reports were from newspapers, government reports etc. and it would be useful to have more information on the Propluvia French data as it is not clear what it is. Please include URLs to the sources where appropriate.

 $\rightarrow$  Yes, we agree with that and will give further information in the revised manuscript. Propluvia is an official government portal that publishes water use restrictions all over France (http://propluvia.developpement-durable.gouv.fr).

Section 2.2 - I suggest that you could put the example impact reports in a table – this would be easier to read and for readers to see the differences between the impact data from the different sources.

 $\rightarrow$  We see your point and will test if it is improving the text or not.

L149 – the choice of the case study years has not been explained – it would be good to introduce these years in the introduction perhaps with appropriate references in the introduction e.g. Lahaa et al. 2017 (https://doi.org/10.5194/hess-21-3001-2017)

 $\rightarrow$  We agree with you that this needs to be clarified. The selection of these years was based on the impact data we compiled. We simply found substantially more reports for these years, wherefore we wanted to specifically focus on them. See also our answer to the related point below (Discussion section).

**Results**:**

L193 – The percentages here do not match up with those previous sentence – please check these figures

 $\rightarrow$  Thanks for pointing this out. We will correct the mistake in the numbers.

Table 1 – foot note 1 should be moved to the caption for clarity (and could be mentioned in the text). You could consider showing these results in a heat map of all the pairs, highlighting these significant pairs (same for Table 2).

 $\rightarrow$  We agree and will revise the caption of the table. We have a heatmap of all the pairs, but as there are more than 30 NUTS regions this Figure is really large and presents a lot of non-significances. Therefore, we decided to summarize the results in Table 1.

Figure 2a – please add a legend for the grey shading of the NUTS regions and make the country borders clearer

Figure 2b – in the caption please explain that data are shown for each sub-category so it is clear why for example the southern region has two labels for the agriculture impacts and why there are faint grey lines within each impact category block

---

## Author Response (AR1)

**Authors Response**

We would like to thank all three Referees for their constructive comments and feedback on this manuscript. We think that the suggested revisions based on the Referee's comments certainly improved the article. Main improvements were modifying the Figures, clarifying the overlap between EDII$_{EU}$ and EDII$_{ALPS}$, and adding further information in the Discussion. Please find our detailed responses (in blue) to each of the Referees below.

In addition to the Referees' suggestions, we added further information in the Supplements to present the results based on the European Drought Impact report Inventory (EDII) before our update (i.e. status of September 2019). We did not include this as a sensitivity analysis in the main article, as the study goal was not a sensitivity analyses of the EDII database. This way the main article remains to focus on drought impacts across the European Alpine region according to our intended research goals.

During the preparation to publish the EDIIALPS V1.0 dataset we found some minor mistakes in the database that are corrected in the revised version. Therefore, some numbers slightly changed. This had no effect on the main results, discussion points or the conclusions.

**Reply to Referee 1: Anne van Loon**

One comment is on the region you are looking at. You need to be clear from the start about the area you are studying. Now you mention 'Alpine Space', European Alpine region, mountain-foreland region, mountain-to-foothill transitions, etc. 'Alpine Space' is now defined in the Methods section, but it should be explained earlier in the Introduction.

→ We agree that this needs to be more clear. See detailed answers below.

My second point is on the compilation of the database. You need to explain more on how the information from the different sources were collected. Have all these data sources been compiled before and are they publicly available? Or have you collected, translated and mined text-based reports yourself?

→ A mix of both. We agree the source data and its compilation needs clarification comment more on the availability of the original sources below. In addition, we made the specific classified collection that we used here available through a repository.

Thirdly, please explain how you have dealt with overlap between EDIIeu and EDIIalps. Did you include EDIIeu impact reports for your region into EDIIalps? Or did you include your EDIIalps in the EDII database? It would be very interesting to compare your EDIIapls with the original EDII entries for your Alpine region to see the effect of differences in data collection.

→ This comment follows the first one. We clarified both in the revised version.

Finally, I found it surprising that there is not many impacts reported in the Energy category. Please expand the discussion on this. Also in the Discussion, I expected you to say more about how perceptions of drought might have influenced the results, for example for the Energy category and for the impacts in the southern part of the Alpine region.

→ We were surprised as well. There are some possible explanations that we added to the discussion (section 4.1). See details below.

**Specific comments:**

**Abstract:**

What is 'Alpine Space'?

→ The EU Interreg-Programm Alpine Space introduced the spatial extent 'Alpine Space' which "covers the Alps and their foothills, as well as different climatic zones", as we defined in section 2.1. We see that this definition came late in the text and therefore now refer to the study region as "Alpine region" in the abstract.

"The amount of more than 3200 compiled reports on negative drought impacts demonstrates the need to move from emergency actions to better preparedness" > not sure if the amount of impact reports demonstrates this need, maybe rephrase?

→ We agree that the conclusion should incorporate more arguments than solely the amount of negative drought impacts. We rephrased the sentence so that our study demonstrates drought as a hazard that requires attention in the Alpine region (l. 16-20).

**Introduction:**

Socio-economic drought: It is a bit confusing how this is different from the impacts that you are investigating. For example the sentence: "These indirect impacts are the least tangible and often related to DSE." (And in the Methods: "DSE is challenging to relate to specific impacts") Please remove DSE as a drought type to avoid confusion between hazard and impacts?

→ We see the confusion and removed DSE.

P.3 l.58: maybe also mention DH impacts on hydropower production. A quite important sector in the Alpine region I thought. But from your results I see that the Energy and industry category is not often reported. Do you know why? Is it not an issue or does this just not end up in the newspapers?

→ We also expected this category to be more often reported, as it is an important sector in the Alpine region. Reasons (anecdotally suggested by stakeholders) include a stronger dependence on the energy market than on water. Also, most hydropower is produced from reservoir storage and most profitably sold for peak demand, whereas run-off-the-river hydropower sold for base demand financially plays less of a role for the (diversified) producers. We added some points to the discussion section 4.1 (l. 408-411).

**Methods:**

Fig. 1: can you indicate the countries and their borders in the map? This is important as later in Fig. 2 & 3 you report impacts by country.

→ Yes, we modified the maps in Fig. 1 in order to show as well the countries, respectively part of the countries in the Alpine Space. We now show the borders in the additional overview map. Otherwise, the overlapping lines of the countries, NUTS 2 and 3 regions, and domains are not visible separately.

Grouped into four domains: Northern, Southern, high-altitude and pre-Alpine. Please make it clear in the text that these are two different subdivisions. So you actually grouped two times into two domains. This comes back in the Results section (see below).

→ Correct, we see that our phrasing was misleading and we clarified this grouping in the revised version (l. 91-95).

P.4 l. 98: explain how you defined Europe as comparison for the Alpine Space analysis.

→ Please read our response to your following comment.

How did you check for overlap with the existing EDII? Did you include your EDIIalps in the EDII database, or the other way around did you include original EDII entries in your EDIIalps database? Did you compare your EDIIalps with the EDII entries for your Alpine region to see the effect of differences in data collection?

→ We agree that our explanation of how we defined the different regions was not clear enough and thus raised your questions. We rephrased the parts describing how we updated EDII and subsetted the different regions in section 2.2 of the revised version. In addition, we modified Fig. 1 to clarify this as well visually and used the terms EDIIalps and EDIIeu throughout the text.

P. 5 l.106-110: Are all the sources listed publically available? Or did you also compile some of these yourself, for example the Italian and German text-based reports? What is the origin of these text-reports?

→ The data we compiled were all publicly available. They stem from different sources meaning that some of them have been compiled before for other overviews or databases, such as the Unwetterchronik and the DMCSEE bulletins. We classified and transferred them into the EDII system. We collected the Italian and German text-based reports ourselves by searching News print and other media etc. and also classified and entered these into the EDII system. We specified the information sources to keep track of the origin of the text-reports. That means that we have the possibility to trace back, if the original source was a research article, newspaper article, governmental report, an entry in another database and so forth. This way we kept the standards the European Drought Impact Report Inventory (EDII) requires. We added further details throughout section 2.2.

**Results:**

Fig. 2: I would suggest to add the distribution of reported impacts in the Alpine Space but based on the EDIIeu, so that you can compare with those in your EDIIalps. Also, maybe refer to the figure a bit more in the text to help the reader.

→ The distribution of reported impacts in the Alpine Space is based on EDIIeu meaning that EDIIalps is a subset of EDIIeu (see our response in the methods section). We integrated the Figure more in the text, especially in section 3.1.

P. 7 l. 185-186: here you need to be careful with the regions again. You can only compare between the regions within a subdivision, so Northern with Southern and pre-Alpine with high-altitude, so rephrase to "… more reports located in the Northern than in the Southern region and more in the pre-Alpine than in the high-altitude region".

→ We rephrased the sentence according to your suggestion (l. 202-203).

Table 1: Add in the caption that this is a subset / selection of tested combinations. Now this is a bit hidden in the table footnote. In general extend the caption so that it is clear which results are presented in the table.

→ According to your suggestion we rephrased the caption and the footnote.

P.7 l. 191-193: Is there a mistake in the numbers or am I failing to understand the difference between the numbers? For EDIIeu the numbers of the two categories add up to 56%, but for EDIIalps 48 + 21 = 69% and not 61%.

→ Thanks for this hint! There is a mistake in the numbers. We corrected the fraction to 69 %.

P.10 l.38-39: "the extreme relative dominance of the category Agriculture and livestock farming in 1976 (82 %) decreased substantially in the years 2003, …" rephrase to something like "the category Agriculture and livestock farming was extremely dominant in 1976 (82 %) and much less apparent in the years 2003, …", because there is not necessarily a change over time.

→ We rephrased the sentence according to your suggestion (l. 255-256).

P.11 l.256: Alpine Space

→ We corrected the spelling mistake.

Table 2: Like in Table 1, extend the caption so that it is clear which results are presented in the table.

→ According to your suggestion we rephrased the caption the footnote.

Figure 5: In the figure, Dsm and Dh are represented as SMD and HD. Please change for consistency and clarity.

→ Thanks for pointing that out. We changed the labels in Figure 5 accordingly.

**Discussion:**

P.15 l.346: "30 % of all impact data across Europe is located in our study region" > Do you mean that within EDIIeu 30% of the impact report is located in your Alpine Space? Or that your EDIIalps has 30% of the impact reports of EDIIeu? The latter is not a conclusion you can draw because of the differences in data collection.

→ We mean that 30 % of all impact data in EDIIeu is located in the Alpine Space which is covered by the EDIIalps (see our explanation to your question in the method part).

P.16 l.378: "ressort"? Do you mean report?

→ We did not mean "report", but the federal forest institutions in Germany that regularly publish assessments about the forest conditions. We will rephrase the sentence to mitigate misunderstandings with the word "ressort" (l. 403).

Please discuss the relatively low amount of impacts in the category Energy (and Tourism).

→ We also expected the categories Energy and tourism to be more reported. For the energy sector, please see our comments above about the dominating role of energy price and market as well as the role of reservoir storages. In general, drought is rarely associated with cold winter droughts, but Alpine tourism is mainly affected by winter droughts. This could be the reason for the relatively low amount of impacts. In fact, our project group also expected tourism to be more present in reports. There may be reasons such as an overlap low economic relevance nationally or in those places where reports are made, the language used to refer e.g. to economic losses due to low snowpack for winter sports and a lack of a verbal link to "drought" (and related terms), which the EDII guidelines require to be made by a report used to create an impact entry. Nevertheless, impacts on *Tourism and recreation* peak in winter, driven by the high-altitude region, as we discussed in section 4.3 (l. 492-496).

P.17 l.413-414: Or maybe because dryness is more normalised in a Mediterranean climate and water shortages are not always reported as drought impacts?

→ This might be an explanation we added in the revised version. As this point better fits to the impacts' spatial heterogeneity across the study region. Therefore, we rearranged this point to section 4.1

P.19 l.453-455: Or water shortages because of a delayed or lower snow melt might not be reported as drought impacts?

→ We think your raised point is especially important in the mountainous regions. Not only are some drought impacts delayed in the Alpine region, they also can occur in another region (upstream, downstream) and therefore might not be associated with drought. We elaborated this point in section 4.1 of the revised manuscript.

**Conclusions:**

The first paragraph (l.475-487) fits better in the Discussion section than in the Conclusions.
→ We also had this discussion beforehand and integrated the paragraph to different sections in the discussion.

**Reply to Referee 2**

Both the data retrieval and the characteristics of the dataset need to be described further. For instance, the period covered by the dataset should be clearly stated in the text and the relationship between the EU and the Alps dataset in terms of counts and repeated entries should be explicitly addressed. I agree with Reviewer 1 that it would be interesting to describe and discuss the difference found for the overlap region (i.e. Alpine NUTS in the EDIIeu dataset and the EDII Alps dataset) as a result of the new impact retrieval process.
→ Corresponding to our answer to Anne van Loon, we agree that this needs to be more clear. We rephrased the parts describing how we updated EDII and subsetted the different regions in section 2.2. In addition, we modified Fig. 1 to clarify this as well visually and used the terms EDIIalps and EDIIeu throughout the text. We added the period covered by our dataset in section 3.2. Regarding the effect of our impact retrieval process we reran the analysis with a previous version of the database (i.e. status from September 2019) and provided it as additional information in the Supplementary material.
It would be useful to strengthen the discussion in terms of the policy relevance of the database and of the trends that emerged from the data analysis. How could policy makers use them? What type of decisions could they inform?
→ Foremost we consider our contribution research work that will have to be analyzed and adapted further to be used by practitioners and/or policy makers. However we added a more elaborated about ways towards applications based on the relevance of our data and findings ideas in the discussion section 4.3 (l. 521-528) and in the conclusions (l. 551-553).
The manuscript presents and analyze a large body of data and it is always challenging to present large datasets and complex patterns. In some paragraphs I had some difficulties following the text. I have noted down the sentences that I found particularly challenging.
→ Thanks for pointing that out. We improved the sentences you noted so that the reader can follow our ideas.

**Specific comments:**

**Introduction:**

p. 2 line 49 Please check the definition of socioeconomic drought. I would expect it to be "insufficient water availability to meet the ordinary demands of society and economic activities" (now it says "inadequate supply of some economic good…").
→ According to the suggestion by Anne van Loon, we removed the drought type DSE.
p.3 Please check the leading questions and may sure that they can be clearly differentiated. The first and the third one seems very similar to me.
→ We rephrased the leading questions (l. 75-79).

**Methods:**

p.3-4 Please specify the altitude ranges used to delineate the different spatial domains and based on what they were defined.
→ On p. 4 line 96-97 we defined the high-altitude region and the pre-Alpine region as follows:
(3) The "high-altitude region" identified with NUTS 3 regions for which ≥ 30 % of the area are higher than 1000 masl versus (4) the "pre-Alpine region" covering all remaining NUTS 3 regions.
p. 5 Please specify in the text the time period covered by the drought impact search. IT would be useful also to know more about the search process: did the authors use a search by key words? If so, what words did they use and how effective the search was?
→ EDII first impact report gets back to 1448. This very historical information for southwestern Germany was retrieved from the collaborative research environment tambora.org (https://www.tambora.org/; Glaser et al., 2015). We added this information on p. 4 l.105-108. However, most collected reports stem from the late 20th century as shown in Fig. 3 and described in section 3.2. We applied the same search method as described in Stahl et al. (2016) in order to be consistent and added this information in our revised manuscript.
p. 6 line 167 "loess": do you mean "loss"?
→ According to the applied method, we mean "loess".
p. 7 line 183: Do the events identified in this new search partially overlap with the EDIIeu ones? How many about of the 3,200 are also counted among the 10,600 ones?
→ As EDIIalps is part of EDIIeu all of the reports within the Alpine Space region are also counted for EDIIeu. See our response to Anne van Loon pasted below:
> We agree that our explanation of how we defined the different regions was not clear enough and thus raised your questions. We rephrased the parts describing how we updated EDII and subsetted the different regions in section 2.2 of the revised version. In addition, we modified Fig. 1 to clarify this as well visually and used the terms EDIIalps and EDIIeu throughout the text.

**Results:**

p. 8 line 225-226: please rephrase the sentence "Thus, the frequency …" (difficult to follow)
→ We agree and rephrased it (l. XX).
p. 9: Could you please explain the usefulness of comparing the NUTS 2 regions (Table 1)? What information does this comparison provide?

→ We wanted to compare not only the national parts of the Alpine Space, but as well smaller regions, as the mountainous terrain is very heterogeneous. Therefore, we included the comparison of the NUTS 2 regions. In Table 1 we show the several NUTS 2 regions located in Italy differed from NUTS 2 regions in Austria, Switzerland and Germany. An effect we did not see in the comparison between the countries.

p.10 lines 238-252 the text is difficult to follow. I recommend simplifying it.

→ We agree and rephrased it (l. 242).

**Discussion:**

p.14 the authors conclude that the chosen data sources proved to be suitable as impacts were clearly liked to the drought occurrence. I suggest revising this statement: the data collection was set up to detect only impacts that are explicitly linked to drought in reports that are being searched, so it is no surprise that the retrieved impacts met that requirements. Instead, the authors could discuss (or at least mention as a caveat/limitation) to what extent they may have missed drought impacts that were not explicitly linked to drought in the reports.

→ We rephrased the sentence and added further information about the limits of our data retrieval in Section 4.1.

p.16 line 364, I think "common" (or similar word) is missing between "most impact".

→ Thanks. We rephrased it to "The second most frequent impact category…" (l. 389)

p.16 line 370-372: please rephrase the sentence starting with "Whether upstream", it is difficult to follow.

→ We agree and rephrased the sentence (l. 394-396).

p.17 line 414: why would impacts in the Southern region be "too local"?

→ We agree that this phrasing is not precise enough and requires clarification. What we meant is that due to the better coverage of the Northern Region with impact reports and hence 'data', we assume a more regionally complete representativeness. In the Southern Region, impact report data is more scarce and hence may have gaps in the spatial representation. We elaborated this further in section 4.1 and removed this phrasing from the original paragraph.

p.18 line 445: the text says that the Southern region reported the most impacts in spring while on the same page, on line 417, it is said "summer and early autumn are the seasons with the most drought impacts in all domains". Please clarify this apparent contradiction.

→ Thanks for pointing that out. We see that this phrasing raises questions. On line 471 we state that "summer and early autumn are the seasons with the most drought impacts in all domains…" supported by Figure 4. The total counts of the reports for each season show that the most impacts were reported in summer and autumn for all plots (a) - (f). If we compare the Southern region with the other regions within the Alpine Space, then this is the region reporting relatively the most (14 % of all reports stem from spring). We rephrased the sentence to clarify this (l. 497).

p.19 line 460 please rephrase (unclear sentence)

→ We agree and rephrased the sentence (l. 513).

p.19 lines 465-466 Please rephrase (unclear sentence)

→ We agree and rephrased the sentence (l. 517).

**Conclusions:**

p.20, line 483: the conclusion "impact data collection EDIIalps is therefore shaped by national priorities and societal effects "is unclear. Also the recommendation about customization EDII (lines 485-487) require some more elaboration in terms of what that "customization" would be.
→ We agree that the sentence is unclear. What we wanted to express is that the data collection might be influenced by the different national foci and different collection efforts. We added this to the discussion on p. 18, l. 412-420.

p.20 Line 488: I recommend to rephrase the sentence starting with "our study…" as the fact of being water rich does not make a place not vulnerable to drought.
→ We agree with you that the natural hazard of drought can occur everywhere, since drought is defined as a deviation from normal. However, drought impacts as an expression of vulnerability or exposure or just as an issue of public awareness are not typically associated with the entire Alpine region. We rephrased the sentence (l. 533).

p.20 line 493-494: please elaborate on the idea of the growing diversity of impacts over time. Is it really due to an increasing complexity of the socioeconomic system in the Alpine Space? Beyond the use of water to produce snow in ski resort, I would expect all the other uses and sectors affected by impacts in 2018 to exist and be well established also in the 1970s and later.
→ We agree that the increase in diversity of impacts may warrant a bit of weighing of possible causes for this observation that in fact we cannot disentangle easily. One aspect which influences the database is the increase in information and access to information in general and this will be reflected in the breadth of impacts. A real growth in diversity of impacts over time, however, can also not be excluded as an explanation. Winter tourism is not the only sector which has changed since 1976. There was also an increase in population, summer tourism, water use etc. In the Alpine forelands the agriculture sector changed a lot and in the Alpine regions new infrastructure for energy and water use was established and subject to more market competition.

p.20 lines 498-501. As it is written now, the reader could think that the authors have compared the impact patterns with actual precipitation patterns or drought indices. It is my understanding that this comparison is beyond the scope of the paper. Instead, in section 4.3 the authors made an interesting attempt to explain the occurrence of impacts throughout a generic year based on the literature. I recommend rephrasing these lines to make sure that they reflect the actual content of paper's analysis.
→ We agree and changed the sentence to: "For the mountainous regions, we could demonstrate the delay between impacts classified as related to soil-moisture drought and those classified as related to hydrological drought." (l. 546-547)

p.20, line 501-02. Please rephrase the sentence "all these…starting point" (unclear)
→ We agree and rephrased the sentence (l. 548).

**Reply to Referee 3**

**Methods:**

Figure 1 – adding the country boundaries would be helpful here especially as you use the countries in the analysis later on. In the right hand figure there seems to be a small region inside the high altitude region that isn't labelled. It is also quite hard to see the NUTS boundaries (which are also not labelled – you could consider adding them to the key) it might help to make the maps bigger and the boundaries thicker so they stand out against the elevation.
→ We improved Figure 1 and followed your and Anne van Loons suggestions. We added a better schematic overview to clarify the different domains covered by EDIIeu and EDIIalps. We would like to avoid adding the label of all NUTS regions, as the maps get very noisy and less clean.
Did you consider splitting (or aggregating the NUTS regions, perhaps NUTS3) by basin – this could be interesting in Section 3.1 where you look at the spatial differences in impacts, as well as Section 3.4 where you consider the different drought types.
→ Yes, we also played with this idea and currently have a Master thesis working on this topic. However, in the preliminary results of this thesis we see differences between the basins of Rhine, Danube, Po, and Rhone, which show similar patterns as our results for the Alpine countries in Fig. 2. Due to the additional methodological steps, but more or less similar results, we decided not to include this in the manuscript.
Section 2.2 – further information on the specific sources of impact data for the Alps EDII would be useful. It's not clear for example, whether the Italian and German text reports were from newspapers, government reports etc. and it would be useful to have more information on the Propluvia French data as it is not clear what it is. Please include URLs to the sources where appropriate.
→ Yes, we agree with that and gave further information in the revised manuscript.
Section 2.2 - I suggest that you could put the example impact reports in a table – this would be easier to read and for readers to see the differences between the impact data from the different sources.
→ As we mentioned only few examples and publish the dataset itself, we want avoid an additional table that can be found in the raw data itself. Therefore, we kept the examples in the text.
L149 – the choice of the case study years has not been explained – it would be good to introduce these years in the introduction perhaps with appropriate references in the introduction e.g. Lahaa et al. 2017 (https://doi.org/10.5194/hess-21-3001-2017)
→ We agree with you that this needs to be clarified. The selection of these years was based on the impact data we compile and clarified this in section 2.2, l. 159-160. See also our answer to the related point below (discussion section).

**Results:**

L193 – The percentages here do not match up with those previous sentence – please check these figures
→ Thanks for pointing this out. We corrected the mistake in the numbers.
Table 1 – footnote 1 should be moved to the caption for clarity (and could be mentioned in the text). You could consider showing these results in a heat map of all the pairs, highlighting these significant pairs (same for Table 2).
→ We agree and revised the caption and the footnote of the table. We have a heatmap of all the pairs, but as there are 35 NUTS regions this Figure is large and presents a lot of non-significances. Therefore, we decided to summarize the results in Table 1.

Figure 2a – please add a legend for the grey shading of the NUTS regions and make the country borders clearer

Figure 2b – in the caption please explain that data are shown for each sub-category so it is clear why for example the southern region has two labels for the agriculture impacts and why there are faint grey lines within each impact category block

→ We modified Figure 2 according to your suggestions.

Figure 3 – it could be the resolution of the figure in this draft version, but the red text is quite hard to read – it is also not explained what the p value is in the caption.

Figure 3 – I guess that the dotty plot is the 'counts of all reports per country and year' - please add a legend to indicate what size of the circles mean. I also suggest you label this figure 3b and the current figure 3b, to 3c

→ We modified Figure 3 according to your suggestions.

Table 2 – I assume the dashes in the rows for the high altitude and southern regions indicate that there were no significant pairs for these regions, please clarify this in the caption.

→ We agree and revised the caption and the footnotes of the table.

Figure 4 – the dates for the seasonal summaries start in March for spring, you could consider doing the same for the time series plots so that the values for the winter are all together.

→ We tested this and found no additional patterns and therefore did not include this analysis in the revised version.

Figure 5 – the caption uses the acronyms DSM and DM but in the plot these are labelled as SMD and HD, please make these consistent here (and throughout the paper – sometimes you use the acronym and sometimes not).

→ Thanks for pointing that out. We improved this in the revised version.

Figures 4 & 5 – In the captions for these two figues the sentence "Monthly values are related to frequency of the month with most impacts." isn't very clear - does this mean that for each impact category the monthly data points for each month are from the year with the most impacts in that category? Please clarify this.

→ We agree and clarified the caption of Figure 4 and 5.

**Discussion:**

L387-389 – Here you say that 2003, 2015 and 2018 depicted are more comparable picture, but in the following state there was a remarkable difference in 2003. These points seem to contradict each other – unless the comparable picture refers only to the Agriculture and Forestry categories. Please clarify.

→ We agree that this part is not clearly written. We restructured the whole section 4.2 in order to clarify the differences between the drought events also considering drought literature (see our response to your following suggestion)

Section 4.3 – it would be interesting to consider the temporal trends and drought occurrence in the context of other known drought events (aside from the case study events used in the paper) for example, comparing the results to papers that consider the timing, propagation and characteristics of drought events (e.g. Laaha and Van Loon 2015 (https://doi.org/10.1016/j.jhydrol.2014.10.059), Haslinger & Bloschl 2017 (https://doi.org/10.1002/2017WR020797) such as Spinoni et al 2015 (https://doi.org/10.1016/j.ejrh.2015.01.001), Sheffield et al. 2009 (https://doi.org/10.1175/2008jcli2722.1) and others)

→ We acknowledge the need to better compare our impact data based events with the literature. We elaborated this further throughout Section 4.2 and integrated some of your proposed references and further literature.

**Conclusion:**

You make an interesting comment in the final line of the paper on the application of the EDIIalps data, it would be interesting to explore this further in the discussion.
→ We described further, how EDIIalps might be used in practice such as risk assessments or monitoring in section 4.3.

**English and Minor points**

→ Thank you very much for pointing us to several minor suggestions improving the clarity and structure of the manuscript as well as the written English. We addressed all of them. As well, we added the DOI referring to the used data of the EDIIALPS V1.0.

**References**

Glaser, R., Riemann, D., Kellersohn, A., Lentz, S., Hanewinkel, C., Beck, A., Vogt, S., Borel, F., Sidawi, W., Kahle, M., Vogt, J., Steller, H., Specht, S., and Koslitz, S.: Tambora – the climate and environmental history collaborative research environment, FreiDok plus, Albert-Ludwigs-Universität Freiburg, Freiburg, https://doi.org/10.6094/tambora.org, 2015.

Stahl, K., Kohn, I., Blauhut, V., Urquijo, J., De Stefano, L., Acácio, V., Dias, S., Stagge, J. H., Tallaksen, L. M., Kampragou, E., van Loon, A. F., Barker, L. J., Melsen, L. A., Bifulco, C., Musolino, D., de Carli, A., Massarutto, A., Assimacopoulos, D., and van Lanen, H. A. J.: Impacts of European drought events: insights from an international database of text-based reports, Nat. Hazards Earth Syst. Sci., 16, 801–819, https://doi.org/10.5194/nhess-16-801-2016, 2016.